# Certified Robustness under Heterogeneous Perturbations via Hybrid Randomized Smoothing

Blaise Delattre [1]   Hengyu Wu [1]   Paul Caillon [2]   Wei Yang Bryan Lim [3]   Yang Cao [1]

## Abstract

Randomized smoothing provides strong, model-agnostic robustness certificates, but existing guarantees are limited to single modalities, treating continuous and discrete inputs in isolation. This limitation becomes critical in multimodal models, where decisions depend on cross-modal semantics and adversaries can jointly perturb heterogeneous inputs, rendering unimodal certificates insufficient. We introduce a unified randomized smoothing framework for mixed discrete–continuous inputs based on an analytically tractable Neyman–Pearson formulation of the joint worst-case problem. By analyzing the joint likelihood ordering induced by factorized discrete and continuous noise, our approach yields a closed-form, one-dimensional certificate that strictly generalizes both Gaussian (image-only) and discrete (text-only) randomized smoothing. We validate the framework on multimodal safety filtering, providing, to our knowledge, the first model-agnostic Neyman–Pearson certificate for joint discrete-token and continuous-image perturbations in interaction-dependent text–image safety filtering.

## 1. Introduction

Randomized smoothing (RS) provides strong, model-agnostic robustness certificates by certifying the stability of a *smoothed* predictor under input perturbations (Cohen et al., 2019; Salman et al., 2019). Its theoretical guarantees, however, are fundamentally derived under *homogeneous* perturbation models, where adversarial uncertainty is confined

to a single modality and induces a single likelihood-ratio ordering. As a result, existing RS theory treats continuous and discrete inputs (Ye et al., 2020; Zeng et al., 2023) in isolation and offers no principled way to reason about adversaries that act simultaneously on heterogeneous input channels.

This limitation has become critical in modern learning systems, where decisions increasingly depend on *joint* reasoning over mixed discrete–continuous inputs (Xu & Fu, 2025; Wu & Cao, 2025). In such settings, adversarial perturbations do not merely act on multiple modalities, but interact through their combined semantic interpretation (Yin et al., 2023). Certifying each modality independently or collapsing heterogeneous perturbations into a single norm fails to capture this joint worst-case behavior. The missing theoretical ingredient is therefore not joint perturbations per se, but a principled *joint likelihood ordering* under heterogeneous noise models, which classical randomized smoothing does not provide.

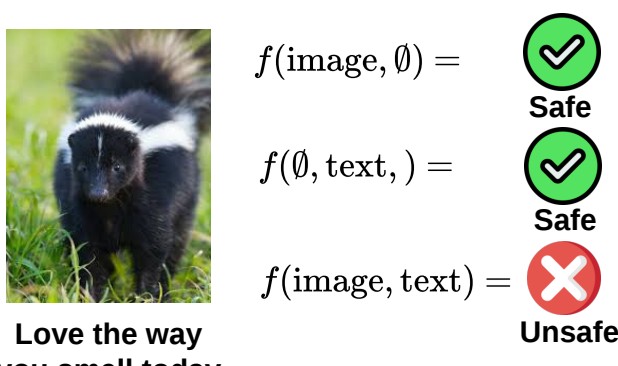

Love the way you smell today

*Figure 1.* Conceptual interaction-only safety behavior. Each modality is classified as Safe in isolation, while the joint text–image input is classified as Unsafe. This illustrates why the certified object must be the joint multimodal decision rather than two unimodal decisions.

Multimodal safety filtering in large foundation models provides a canonical and high-stakes instance of this problem. Large multimodal models are increasingly deployed in safety-critical domains such as medical imaging (Nam et al., 2025), robotics (Kawaharazuka et al., 2025), and autonomous systems (Cui et al., 2024), where formal

*Equal contribution [1]Department of Computer Science, School of Computing, Institute of Science Tokyo, Tokyo, Japan [2]PSL University, Paris, France [3]College of Computing and Data Science, Nanyang Technological University, Singapore. Correspondence to: Blaise Delattre <delattre.b.aa@m.titech.ac.jp>.

*Proceedings of the 43rd International Conference on Machine Learning*, Seoul, South Korea. PMLR 306, 2026. Copyright 2026 by the author(s).

guarantees are required rather than empirical robustness alone. Their open-ended interfaces expose them to *prompt-injection attacks* that manipulate both discrete prompt tokens and continuous perceptual representations (Clusmann et al., 2025; Carlini et al., 2023a; Zou et al., 2023; Liu et al., 2024). Crucially, safety violations in vision–language systems are often *interaction-level* phenomena: neither the image nor the text is unsafe in isolation, while their composition is. This makes unimodal certification intrinsically insufficient, as the object of interest is the stability of the *joint* multimodal decision (Figure 1).

We develop a modality-agnostic randomized smoothing framework for heterogeneous perturbations, designed to certify robustness under joint discrete and continuous adversarial actions. Multimodal safety provides a motivating and practically relevant instance, without restricting the scope of the framework. Within this setting, we introduce a unified hybrid randomized smoothing formulation and characterize the joint worst-case behavior of the smoothed classifier through a Neyman–Pearson analysis. A key insight is that the presence of a continuous component fundamentally changes the structure of the discrete worst-case problem: purely discrete smoothing yields non-invertible likelihood orderings, while continuity restores a tractable one-dimensional characterization of the joint robustness trade-off. Our contributions are as follows:

- **Hybrid Neyman–Pearson theory for heterogeneous perturbations.** We derive, in Section 4.1, an exact Neyman–Pearson characterization for joint discrete–continuous randomized smoothing, yielding a one-dimensional, continuous, and invertible likelihood-ratio CDF that strictly generalizes both discrete knapsack-based and Gaussian randomized smoothing.

- **Provably sound hybrid certification procedure.** In Section 4.2, we instantiate the theoretical characterization as a practical certification algorithm with guaranteed monotonicity and conservative numerical guarantees, based on one-dimensional root finding and worst-case aggregation over discrete attacks.

- **Empirical validation in multimodal and mixed-input regimes.** We empirically validate the proposed framework in two complementary settings. First, we study certified multimodal AI safety under joint text–image perturbations (Section 5.2), and assess the tightness of the certificates using adaptive empirical attacks (Section 5.3), on a scoped interaction-only evaluation split where each modality is individually safe but jointly unsafe. Second, we consider a mixed-feature tabular classification task, demonstrating that the hybrid certificate applies beyond vision–language models (Section 5.1).

**Conflict of Interest Disclosure.** The authors declare no financial or other substantive conflicts of interest that could reasonably be perceived to influence this work.

## 2. Related Works

**Prompt Injection and Empirical Defense.** Prompt injection exploits the inability of LLMs to reliably distinguish trusted instructions from untrusted input, allowing adversaries to override the intended task via malicious prompt modifications. Liu et al. (2024) provide an early formal treatment and a unified benchmark covering multiple attack and defense settings. Subsequent work shows that prompt injection can be *universal* and transferable. In particular, Zou et al. (2023) demonstrate that a single adversarial suffix can bypass safety alignment across prompts and models, including black-box systems, exposing fundamental weaknesses in current guardrails. Proposed defenses range from architectural separation, such as structured queries that isolate instructions from data (Chen et al., 2025b), to system-level mechanisms like capability-based control in LLM agents (Debenedetti et al., 2025), and test-time input transformations such as DefensiveTokens (Chen et al., 2025c). However, many of these defenses degrade under adaptive attackers (Zhan et al., 2025), remain vulnerable in cross-lingual settings (Yong et al., 2023), or lack formal guarantees (Khomsky et al., 2024). These limitations motivate certified approaches that seek provable guarantees against prompt injection rather than empirical defenses alone.

**Safety Detectors for Certified Robustness to Prompt Injection.** Kumar et al. (2024) frame prompt-injection robustness within a model-agnostic randomized smoothing perspective. Chen et al. (2025a) further derive tight worst-case certificates for randomized LLM safety defenses via fractional and 0–1 knapsack solvers. These works certify discrete prompt perturbations, whereas we certify heterogeneous attacks combining discrete token changes and continuous $\ell_2$ image perturbations under a single joint Neyman–Pearson analysis.

**Multimodal Adversarial Attacks and Defenses.** Prior work shows that jointly optimized multimodal attacks are stronger than unimodal ones, as they exploit cross-modal alignments rather than independent modality weaknesses. Co-Attack demonstrates that perturbing image or text alone is often ineffective, while coordinated perturbations significantly increase attack success on vision–language models (Zhang et al., 2022). VLAttack further confirms that breaking image–text correlation typically requires joint optimization (Yin et al., 2023). Subsequent work highlights additional amplification mechanisms, including set-level alignment exploitation and collaborative interaction attacks such as CrossFire (Lu et al., 2023; Dou et al., 2024).

Existing certified defenses remain limited. Randomized smoothing has been applied to VLMs by treating them effectively as vision models, with text handled in embedding space (Seferis et al., 2025; Nirala et al., 2024). COMMIT certifies multi-sensor fusion systems via shared-latent continuous noise, targeting geometric transformations rather than heterogeneous text–image perturbations, and is not a classical randomized smoothing approach (Huang et al., 2025). CertTA provides an ad hoc certification method for traffic networks based on composed kernels (Yan et al., 2025), following a similar underlying principle to worst-case robustness analyses for LLMs (Chen et al., 2025a).

**Comparison with Multimodal Certified Robustness.** MMCert (Wang et al., 2024) gives a certified defense for multimodal models by independently subsampling basic elements from each modality and aggregating predictions. Its threat model is $\ell_0$-like across modalities: the attacker may add, delete, or modify a bounded number of basic elements in each modality. This differs from our setting, where the perturbation geometry is heterogeneous, with token-level discrete perturbations and continuous $\ell_2$ image perturbations. Knowledge Continuity (Sun et al., 2024) instead provides probabilistic robustness guarantees through a representation-space continuity measure that is designed to be domain-independent. This is complementary to randomized smoothing: it is not a worst-case Neyman–Pearson certificate over an explicit joint $(d, \epsilon)$ perturbation budget. These methods address related multimodal certification goals, but their guarantees do not directly instantiate the heterogeneous discrete–continuous threat model studied here. An empirical comparison with an MMCert-style subsampling baseline on our interaction-only Hateful Memes evaluation is provided in Appendix A.10: under independent subsampling at any keep ratio, the smoothed unsafe probability collapses below the certification threshold and yields zero certifiable examples on this subset.

**Randomized Smoothing.** Randomized smoothing (RS) certifies robustness by replacing a base predictor with its noisy expectation. In continuous settings, Gaussian smoothing yields closed-form $\ell_2$ guarantees via a Neyman–Pearson (NP) analysis of shifted Gaussians, exploiting the monotone likelihood ratio of the noise channel (Cohen et al., 2019; Salman et al., 2019; Lecuyer et al., 2019; Li et al., 2018), with extensions to alternative norms and noise distributions (Levine & Feizi, 2020; 2021). For discrete inputs, RS has been applied to text through stochastic kernels such as synonym substitutions, character noise, or masking (Jia & Liang, 2019; Ye et al., 2020; Zeng et al., 2023), producing nontrivial $\ell_0$ certificates. However, unlike the continuous case, discrete likelihood ratios are atomic and induce non-invertible NP decision rules, precluding closed-form certificates. Recent worst-case analyses formalize this gap and show that certification reduces to a combinatorial or-

dering over likelihood ratios rather than a one-dimensional inversion (Lee et al., 2019; Chen et al., 2025a).

As a result, existing RS theory treats continuous and discrete perturbations in isolation and offers no unified certification framework for heterogeneous noise models.

## 3. From Neyman–Pearson to Knapsack: What Breaks in Hybrid Inputs.

We briefly recall a standard fractional-knapsack formulation of randomized smoothing, which serves as a technical starting point. All results up to Eq. (1) follow existing worst-case analyses and are included solely to make the hybrid extension self-contained.

**Certified Robustness and Randomized Smoothing.** We consider binary classification over an input space $\mathcal{X}$, consisting of either continuous components (e.g. images) or discrete components (e.g. text). A classifier $f : \mathcal{X} \to \{0, 1\}$ maps inputs to binary decisions. For illustration, $f(x) = 1$ denotes that the input is flagged as unsafe. $f$ is robust at $x$ up to radius $\delta$ if $f(x_{\text{adv}}) = f(x)$ for all $x_{\text{adv}}$ such that $D(x, x_{\text{adv}}) \leq \delta$, where $D$ is a task-specific perturbation metric. The certified radius is

$$r_{\text{cert}}(x) = \sup\{\delta \geq 0 : \forall x_{\text{adv}}, \ D(x, x_{\text{adv}}) \leq \delta, \ f(x_{\text{adv}}) = f(x)\}.$$

Randomized smoothing (Cohen et al., 2019) constructs a smoothed classifier $g(x) = \mathbb{E}_{z \sim p(\cdot | x)}[f(z)]$, where $p(z \mid x)$ is a noise channel centered at $x$. For binary classifiers, $g(x)$ equals the probability of the positive class. Let $p_A = g(x)$ denote this clean value, estimated via Monte Carlo sampling. For a perturbation budget $\delta$, define the worst-case smoothed value

$$p_{\text{adv}}(\delta) = \inf_{D(x, x_{\text{adv}}) \leq \delta} g(x_{\text{adv}}).$$

For a decision threshold $\tau \in (0, 1)$, define the randomized-smoothing certified radius as

$$r_{\text{RS}}(x; \tau) = \sup\{\delta \geq 0 : p_{\text{adv}}(\delta) > \tau\}.$$

The classical majority-voting certificate corresponds to $\tau = \frac{1}{2}$. Here, we focus on certifying "Unsafe" predictions, requiring the smoothed score to remain above a small threshold $\tau \ll \frac{1}{2}$ under worst-case perturbations. The parameter $\tau$ thus controls the strictness of certification: smaller values yield larger certified radii but tolerate weaker confidence, while larger values enforce stricter guarantees at the cost of reduced coverage, see Appendix A.1 for more details.

**Neyman–Pearson Structure of the Worst Case.** Computing $p_{\text{adv}}(\delta)$ amounts to characterizing the minimal expectation of $f$ under the adversarial channel $p(\cdot \mid x_{\text{adv}})$ subject to a fixed expectation under $p(\cdot \mid x)$. To characterize this worst case, we temporarily relax the discrete classifier $f$

to an arbitrary measurable function $h : \mathcal{X} \to [0, 1]$ with fixed expectation under $p(\cdot \mid x)$, which serves solely as a proof device. Let $\gamma(z) = p(z \mid x_{\mathrm{adv}})/p(z \mid x)$ denote the likelihood ratio. The Neyman–Pearson lemma characterizes the extremal solution.

**Lemma 3.1** (Neyman–Pearson for randomized smoothing (Neyman & Pearson, 1933)). *Among all measurable* $h : \mathcal{X} \to [0, 1]$ *satisfying* $\mathbb{E}_{z \sim p(\cdot \mid x)}[h(z)] = p_A$, *the minimizer of* $\mathbb{E}_{z \sim p(\cdot \mid x_{\mathrm{adv}})}[h(z)]$ *is*

$$h^{\star}(z) = \mathbf{1}_{\{\gamma(z) \leq t^{\star}\}},$$

*where* $t^{\star}$ *enforces* $\mathbb{E}_{z \sim p(\cdot \mid x)}[h^{\star}(z)] = p_A$.

In the continuous Gaussian setting, this yields the closed-form bound $p_{\mathrm{adv}}(\delta) = \Phi(\Phi^{-1}(p_A) - \delta/\sigma)$ and, for threshold $\tau$, the bound on $r_{\mathrm{cert}}$,

$$r_{\mathrm{RS}}(x; \tau) = \sigma(\Phi^{-1}(p_A) - \Phi^{-1}(\tau))$$

, with $\sigma^2$ the smoothing variance and $\Phi$ the Gaussian cumulative distribution function.

**Fractional-Knapsack Formulation for Discrete Inputs.** For discrete channels (e.g. text), the likelihood ratio $\gamma$ has atoms, so the Neyman–Pearson constraint in Lemma 3.1 cannot, in general, be met by a binary threshold rule. The optimal solution may therefore require fractional mass at a likelihood-ratio tie (i.e., $\mathbb{P}_{Z \sim p(\cdot \mid x)}(\gamma(Z) = t) > 0$ for some $t$), leading to the convex program

$$\min_{0 \leq h(z) \leq 1} \quad \sum_z h(z) \, p(z \mid x_{\mathrm{adv}})$$
$$\text{s.t.} \quad \sum_z h(z) \, p(z \mid x) = p_A, \quad (1)$$

which is the likelihood-ratio ordering problem underlying tight randomized-smoothing certificates for discrete spaces (Lee et al., 2019). We refer to this equivalent optimization as a *fractional knapsack* following the terminology of Chen et al. (2025a). Its solution yields the tightest computable lower bound on $p_{\mathrm{adv}}(\delta)$ obtainable from the Neyman–Pearson relaxation. Under exact kernel symmetry, this knapsack reduces to a grouped form, yielding the tightest bound induced by the relaxation.

The hybrid results below are self-contained: Theorem 4.2 uses only the Neyman–Pearson lemma, the factorized discrete–Gaussian likelihood ratio, and elementary Gaussian monotonicity.

**Limitations of Existing Theory.** Up to Eq. (1), the formulation is standard. However, existing randomized smoothing frameworks treat continuous and discrete inputs separately. Gaussian smoothing applies only to continuous spaces, while discrete smoothing induces a non-invertible Neyman–Pearson map that cannot be coupled with continuous noise. As a consequence, no existing theory provides

a principled joint likelihood ordering for mixed discrete–continuous perturbations. Addressing this gap is the focus of the remainder of this paper.

## 4. Hybrid Randomized Smoothing

We now extend randomized smoothing to hybrid inputs combining discrete and continuous components. Let $x = (x_1, x_2)$, where $x_1 \in \mathcal{X}_1$ is discrete and $x_2 \in \mathbb{R}^D$ is continuous. We assume independent smoothing kernels:

$$Z_1 \sim p_1(\cdot \mid x_1), \qquad Z_2 \sim \mathcal{N}(x_2, \sigma^2 I),$$

with joint distribution $p(z \mid x) = p_1(z_1 \mid x_1) \, p_2(z_2 \mid x_2)$, and define the smoothed score $g(x) = \mathbb{E}[f(Z_1, Z_2)]$.

To quantify certified robustness under mixed perturbations, we consider hybrid budgets $(d, \epsilon)$ for the discrete and continuous components, respectively. The ideal robustness certificate is the maximal continuous perturbation radius that preserves the decision under a discrete budget $d$:

$$r_{\mathrm{cert}}(x; d) = \sup \left\{ \epsilon \geq 0 : \forall x_{\mathrm{adv}}, \begin{array}{l} D_1(x_1, x_{1,\mathrm{adv}}) \leq d, \\ \|x_2 - x_{2,\mathrm{adv}}\|_2 \leq \epsilon, \\ f(x_{\mathrm{adv}}) = f(x) \end{array} \right\}.$$

We define the hybrid RS-certified bound on radius:

$$p_{\mathrm{adv}}(d, \epsilon) = \inf_{\substack{D_1(x_1, x_{1,\mathrm{adv}}) \leq d \\ \|x_2 - x_{2,\mathrm{adv}}\|_2 \leq \epsilon}} g(x_{\mathrm{adv}}),$$

$$r_{\mathrm{RS}}(x; d, \tau) = \sup\{\epsilon \geq 0 : p_{\mathrm{adv}}(d, \epsilon) > \tau\}.$$

We define the marginal likelihood ratios:

$$\gamma_1(z_1) = \frac{p_1(z_1 \mid x_{1,\mathrm{adv}})}{p_1(z_1 \mid x_1)}, \qquad \gamma_2(z_2) = \frac{p_2(z_2 \mid x_{2,\mathrm{adv}})}{p_2(z_2 \mid x_2)}.$$

**Why Naive Multimodal Composition Fails.** A natural baseline is to certify the discrete and continuous components independently and combine the resulting guarantees. This is invalid in general. Randomized smoothing robustness is governed by a *single* NP inner problem over $h(z_1, z_2)$ under a single expectation constraint, which does not decompose across modalities. Even when the smoothing kernel factorizes, the NP-optimal acceptance region on the product space need not be separable.

**Proposition 4.1** (Non-composability of NP rules). *Let* $p(z \mid x) = p_1(z_1 \mid x_1) \, p_2(z_2 \mid x_2)$. *In general, the NP-optimal acceptance region on* $\mathcal{X}_1 \times R^D$ *cannot be written as* $\{\gamma_1(z_1) \leq t_1\} \cap \{\gamma_2(z_2) \leq t_2\}$ *for any thresholds* $t_1, t_2$.

**Counterexample (Discrete $\times$ Continuous).** Let $z_1 \in \{a, b\}$ with likelihood ratios $\gamma_1(a) = 1$ and $\gamma_1(b) = M \gg 1$, and let $z_2$ follow a Gaussian channel with shift $r > 0$. The joint likelihood ratio satisfies $\log \gamma(z_1, z_2) =$

$\log \gamma_1(z_1) + r z_2 - \frac{r^2}{2}$. The NP-optimal acceptance region is a half-space in the joint variable $\log \gamma_1(z_1) + r z_2$, intrinsically coupling the discrete and continuous coordinates. No choice of separate unimodal thresholds can recover this region; equivalently, perturbations may satisfy each unimodal certified radius while violating the joint NP constraint, so unimodal certificates do not compose.

### 4.1. Hybrid Neyman–Pearson Certificate

The following theorem shows that the hybrid NP inner problem admits a closed-form, one-dimensional characterization even under heterogeneous discrete–continuous perturbations. The central object is a *likelihood-ratio CDF* $F(t; r)$ that quantifies how much probability mass can be accepted under the NP constraint at a given continuous radius. Crucially, this map is continuous and strictly increasing, so the optimal hybrid test is governed by a *single* scalar threshold.

**Theorem 4.2** (Hybrid randomized smoothing certificate).
*Fix* $x = (x_1, x_2)$ *with clean smoothed value* $p_A = g(x) \in (0, 1)$. *Let* $x_{\mathrm{adv}} = (x_{1,\mathrm{adv}}, x_{2,\mathrm{adv}})$ *satisfy* $D_1(x_1, x_{1,\mathrm{adv}}) \le d$ *and* $\|x_{2,\mathrm{adv}} - x_2\|_2 \le \epsilon$, *and set* $r = \|x_{2,\mathrm{adv}} - x_2\|_2$. *Define*

$$F(t; r) = \sum_{z_1} p_1(z_1 \mid x_1) \, \Phi\left( \frac{\frac{r^2}{2} + \sigma^2 (\log t - \log \gamma_1(z_1))}{\sigma r} \right),$$

*where* $\Phi$ *is the Gaussian cdf. For each* $r > 0$, *there exists a unique* $t^\star(r) > 0$ *such that* $F(t^\star(r); r) = p_A$.

*The Neyman–Pearson optimal rule for the hybrid problem is the likelihood-ratio test*

$$h^\star(z_1, z_2) = \mathbf{1}\{\gamma(z_1, z_2) \le t^\star(r)\}.$$

*The resulting worst-case smoothed probability at radius* $r$ *is*

$$V(x_{1,\mathrm{adv}}; r) = \sum_{z_1} p_1(z_1 \mid x_{1,\mathrm{adv}}) \, \Phi\left( \frac{\frac{r^2}{2} + \sigma^2 \left( \log t^\star(r) - \log \gamma_1(z_1) \right)}{\sigma r} - \frac{r}{\sigma} \right).$$

*Moreover, for any fixed* $x_{1,\mathrm{adv}}$, $V(x_{1,\mathrm{adv}}; r)$ *is nonincreasing in* $r$, *so the continuous worst case is attained at* $r = \epsilon$. *Taking the worst case over all discrete adversaries with* $D_1(x_1, x_{1,\mathrm{adv}}) \le d$, *the hybrid worst-case smoothed value under budgets* $(d, \epsilon)$ *is*

$$p_{\mathrm{adv}}(d, \epsilon) = \inf_{D_1(x_1, x_{1,\mathrm{adv}}) \le d} V(x_{1,\mathrm{adv}}; \epsilon).$$

See proof in Appendix A.2. Theorem 4.2 provides a closed-form characterization of the worst-case smoothed value under joint discrete–continuous perturbations.

The hybrid threshold $t^\star(r)$ captures the exact NP trade-off at radius $r$, while $V(x_{1,\mathrm{adv}}; r)$ evaluates the corresponding worst-case smoothed probability. Minimizing $V$ over admissible discrete perturbations yields $p_{\mathrm{adv}}(d, \epsilon)$.

**Monotonicity in the Discrete Budget.** For fixed continuous radius $r$, let $\widehat{p}_{\mathrm{adv}}(d, r)$ denote the certified worst-case value at discrete budget $d$. Since adversary sets are nested, $\mathcal{A}(d') \subseteq \mathcal{A}(d)$ for $d' \le d$, we have $\widehat{p}_{\mathrm{adv}}(d', r) \ge \widehat{p}_{\mathrm{adv}}(d, r)$. Consequently, for any $\tau$, certified radii and certified accuracies are pointwise nonincreasing as $d$ increases.

**Continuous Smoothing Regularizes the Discrete Knapsack.** When the input contains only a discrete component, the NP acceptance map is stepwise in the likelihood-ratio threshold and the inner problem reduces to a fractional knapsack with a non-invertible capacity constraint. In the hybrid setting, the Gaussian component introduces a strictly monotone factor in $\gamma(z_1, z_2)$, and the resulting likelihood-ratio CDF $F(t; r)$ becomes continuous and strictly increasing in $t$. Thus the continuous modality smooths the discrete likelihood ratios and regularizes the knapsack structure into an analytically invertible one-dimensional problem.

Here, $\sigma$ plays a dual role: beyond controlling the continuous robustness radius, it regularizes the discrete likelihood ratios by breaking atomic ties, inducing a smooth and invertible relaxation of the discrete NP problem.

**Recovery of Unimodal Limits: Consistency with Continuous- and Discrete-Only Certificates.** The hybrid certificate recovers the classical unimodal guarantees as special cases. If $x_{1,\mathrm{adv}} = x_1$, then $\gamma_1 \equiv 1$ and the capacity equation reduces to a one-dimensional Gaussian inversion, exactly recovering the standard $\ell_2$ randomized smoothing certificate. If there is no continuous perturbation, the NP problem reduces to a purely discrete likelihood-ratio test, yielding the classical fractional-knapsack worst-case bound over $\gamma_1$. A formal derivation of this discrete limit, obtained as $\sigma \to \infty$, is given in Appendix A.3. Thus, the hybrid formulation strictly generalizes both unimodal settings without loss of tightness.

### 4.2. Implementation of the Hybrid Certificate

We describe how to compute a certified lower bound $p_{\mathrm{adv}}(d, \epsilon)$ for a fixed input $x = (x_1, x_2)$ and budgets $(d, \epsilon)$ using the quantities $F$ and $V$ from Theorem 4.2. The procedure reduces the hybrid certificate to a one-dimensional NP root-finding problem for each admissible discrete perturbation.

1. **Estimate the clean smoothed score.** Draw i.i.d. samples $(Z_1^{(n)}, Z_2^{(n)}) \sim p(\cdot \mid x)$ and estimate

$$p_A \approx \frac{1}{N} \sum_{n=1}^{N} f\big(Z_1^{(n)}, Z_2^{(n)}\big).$$

Using a one-sided Clopper–Pearson interval at risk level $\alpha$, we obtain a lower confidence bound $\hat{p}_A$ on the clean smoothed score.

2. **Discrete worst-case aggregation.** Compute the worst-case over all admissible adversarial texts $x_{1,\text{adv}}$ satisfying $D_1(x_1, x_{1,\text{adv}}) \leq d$. For small budgets, this can be done by exact enumeration. For common symmetric text-smoothing kernels (e.g., uniform or absorbing), the induced distribution $p_1(\cdot \mid x_{1,\text{adv}})$ depends only on the budget $d$ and not on the specific adversarial suffix, allowing the discrete worst case to be characterized analytically without explicit enumeration; formal definitions are given in Appendix A.8.

3. **Solve the hybrid Neyman–Pearson constraint.** For each candidate $k$, compute the discrete likelihood ratios $\gamma_1^{(k)}(z_1) = p_1(z_1 \mid x_{1,\text{adv}}^{(k)})/p_1(z_1 \mid x_1)$ and set $r = \epsilon$. Solve the one-dimensional equation $F_k(t) = \hat{p}_A$ for $t_k^\star > 0$ (e.g. by bisection in $u = \log t$), where $F_k$ is defined in Theorem 4.2. Existence and uniqueness follow from monotonicity of $F_k(\cdot)$.

4. **Evaluate the hybrid adversarial value.** Compute the corresponding adversarial score $V_k = V(x_{1,\text{adv}}^{(k)}; r)$, using the closed-form expression from Theorem 4.2.

5. **Aggregate over discrete attacks.** Take the worst case $\hat{p}_{\text{adv}}(d, \epsilon) = \min_k V_k$.

6. **Certification decision.** In the binary case with threshold $\tau$, the prediction at $x$ is certified within budgets $(d, \epsilon)$ if the worst-case adversarial value $\hat{p}_{\text{adv}}(d, \epsilon)$ remains on the same side of $\tau$ as $\hat{p}_A$.

**Structural Symmetry of Text Smoothing Kernels.** We consider standard discrete smoothing kernels, including uniform and absorbing/masking, widely used in prior work (Jin et al., 2020; He et al., 2022; Meng et al., 2022; Lou et al., 2023; Chen et al., 2025a). Each suffix token is independently corrupted with probability $\beta$, which controls the discrete robustness budget. For *uniform* replacement, corrupted tokens are drawn uniformly from the vocabulary; for *absorbing* kernels, they are replaced with a fixed mask token. In both cases, the smoothed distribution $p_1(\cdot \mid x_{1,\text{adv}})$ depends only on the edit budget (suffix length or $\ell_0$ sparsity) and not on token identity. This structural symmetry enables efficient worst-case evaluation across all discrete threat models via a canonical adversarial input, avoiding combinatorial search.

**Degeneracy of Absorbing Kernels for Suffix Attacks.** However, for suffix attacks, absorbing kernels induce degenerate Neyman–Pearson bounds: the likelihood ratio collapses to a two-point distribution, and certification requires $p_A \geq 1 - \beta^d$, with bound at most $\beta^d$, degrading exponentially with $d$. Uniform kernels, by contrast, maintain overlapping support between clean and adversarial distributions, yielding stronger certificates. We adopt uniform

replacement for suffix attacks. Kernel definitions and behaviors are detailed in Appendix A.8.

All numerical steps in the implementation preserve conservative guarantees: the clean smoothed score is lower-bounded via a one-sided Clopper–Pearson interval, the NP threshold $t^\star$ is obtained by monotone root finding, and the outer discrete minimization is handled either exactly by enumeration or conservatively using kernel symmetries, ensuring that the resulting certificate never overestimates robustness. The resulting certificate is inexpensive to compute on CPU relative to model inference and Monte Carlo sampling. Appendix A.7 formalizes numerical precision choices required to preserve conservative guarantees.

# 5. Experiments

We evaluate hybrid randomized smoothing on safety filtering tasks with joint text–image perturbations. Robustness is measured against $\ell_2$ image noise and token-level text attacks. We compare hybrid to unimodal certificates on large pre-trained model, using the uniform kernel with $\alpha = 0.01$, $n = 10^4$, and $\beta = 0.25$ unless stated otherwise.

## 5.1. Hybrid Certification on Tabular Data

This experiment is intended as a sanity check that the hybrid certificate works whenever discrete and continuous features coexist. We evaluate on the ADULT census-income dataset (Becker & Kohavi, 1996; Kohavi, 1996), which combines continuous and categorical features, using a linear SVM trained on standardized continuous variables and one-hot encoded categorical variables. The adversary may apply joint perturbations $\|\delta_{\text{num}}\|_2 \leq \epsilon$, $\|\delta_{\text{cat}}\|_0 \leq d$, where $\|\cdot\|_0$ counts the number of modified categorical fields. For each discrete budget $d$, we report certified accuracy CA. Figure 2 shows a monotone trade-off: certified accuracy degrades as $d$ increases, with no certification beyond $\epsilon = 0.75$ for $d = 2$, while for $d = 0$ certification holds up to $\epsilon = 1.9$ at over 30% CA, confirming that the hybrid certificate captures a joint robustness trade-off.

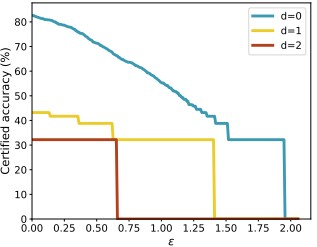

*Figure 2.* Certified accuracy (%) as a function of the continuous $\ell_2$ radius for hybrid randomized smoothing on tabular data, for discrete budgets $d \in \{0, 1, 2\}$.

## 5.2. Certified Multimodal Safety Filtering

We evaluate certified robustness for multimodal safety filtering under joint text–image perturbations using LLaVA-Guard (Helff et al., 2024), a policy-conditioned safety classifier. The model takes as input a triplet $(x_1, x_2, \pi)$, where $x_1$ is a text prompt, $x_2$ is an image, and $\pi$ is a natural language safety policy fixed and not treated as an adversarial or stochastic variable (see Appendix A.6). For all evaluations, we fix $\pi$ to a predefined standard policy describing prohibited content categories. For the image modality, we apply a denoiser-based randomized smoothing scheme, following the denoising-based RS paradigm of Carlini et al. (2023b).

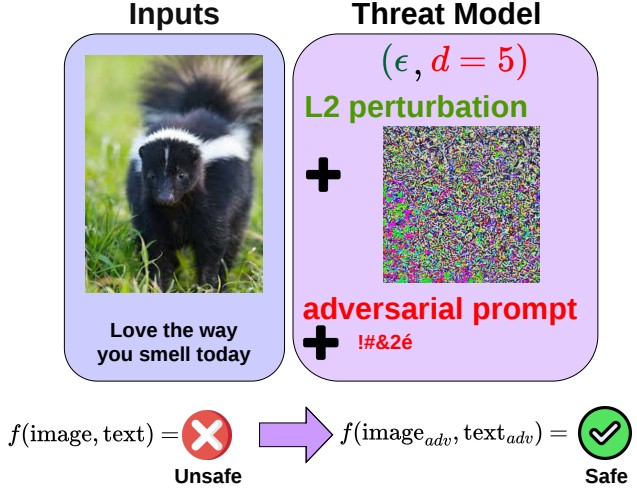

**Figure 3.** Multimodal threat model for safety filtering. The image alone and the text alone are classified as "Safe", while their combination is "Unsafe". An adversary applies an $\ell_2$-bounded perturbation to the image and an append-only suffix attack to the text, with the goal of inducing a "Safe" joint decision.

**Threat Model.** We consider a joint multimodal adversary that perturbs both the text and image inputs.

*Text:* The primary threat is prompt-injection via suffix attacks: starting from a benign prompt $x_1$, the adversary appends up to $d$ tokens to produce $x_{1,\text{adv}}$ with $D_1(x_1, x_{1,\text{adv}}) \leq d$. This covers standard prompt-injection attacks (Zou et al., 2023; Kumar et al., 2024; Chen et al., 2025a). More generally, we handle arbitrary token insertions or replacements, corresponding to $\ell_0$-bounded perturbations, within the same certification framework. Appendix A.4 discusses variable-length inputs.

*Image:* The adversary may apply an $\ell_2$-bounded perturbation $\|x_{2,\text{adv}} - x_2\|_2 \leq \epsilon$.

**Interaction-Only Evaluation.** We focus on an *interaction-only* setting where unsafe behavior emerges only from multimodal composition:

$$f(x_1, \varnothing) = 0, \quad f(\varnothing, x_2) = 0, \quad f(x_1, x_2) = 1.$$

Neither modality alone is sufficient to trigger an "Unsafe" label. Thus, unimodal certification is unsound in this setting (Figure 1).

**Dataset Construction.** We construct interaction-only examples from the Hateful Memes dataset (Kiela et al., 2020). Text-only "Unsafe" prompts are removed using LLaVA-Guard. Remaining meme images have embedded text removed using FLUX.1-Kontext (Labs et al., 2025), and are verified as "Safe" when presented alone. We keep only samples classified as "Unsafe" under the original text-image pair, yielding 400 interaction-only examples. We verify that Gaussian noise alone does not induce "Unsafe" predictions, ensuring that joint behavior drives the unsafe classification.

**Defense to the Multimodal Safety Classifier.** We apply hybrid RS to certify robustness under this threat model. For text, we use a discrete smoothing kernel $p_1(\cdot \mid x_1)$ that corrupts tokens in the suffix region using a fixed noise model (uniform or absorbing). For images, we use Gaussian smoothing with variance $\sigma^2$. The smoothed classifier predicts "Unsafe" when the estimated unsafe probability under the joint distribution exceeds a threshold $\tau$. We fix $\tau = 4.6 \times 10^{-5}$ following Chen et al. (2025a). Since certification is monotone in $\tau$, results are qualitatively stable.

Smoothing reduces clean classifier utility relative to the unsmoothed base model. We report the resulting smoothed accuracy under text and image smoothing in Appendix A.9 (Table 5). Smoothed accuracy degrades mainly through text corruption: under full-text $\ell_0$ smoothing, uniform replacement may alter semantic content throughout the prompt, while under the suffix threat model used for certification, only appended tokens are corrupted, so the core prompt is preserved.

**Comparison to Unimodal Baselines.** We restrict evaluation to examples where at least one method certifies a non-trivial robustness radius (text certification and image–text certification), we get a subset of 100 examples. Table 1 compares certified image and text robustness across baselines. Hybrid RS achieves image radius close to image-only smoothing, with modest reduction from joint certification, and comparable text robustness to text-only RS.

For the suffix attack, hybrid randomized smoothing certifies a joint guarantee $(r_{\text{hybrid}}, d_{\text{hybrid}})$, whereas image-only and text-only methods yield $(r_{\text{img}}, 0)$ and $(0, d_{\text{txt}})$. For image robustness, we report $r_{\text{hybrid}}$ at $d = 1$, the smallest non-trivial text budget, and compare it to $r_{\text{img}}$. Text robustness is compared via $d_{\text{hybrid}}$ and $d_{\text{txt}}$. Average certified values are reported in Table 1. The hybrid certificate yields an image radius of the same order of magnitude as image-only smoothing, with a modest reduction due to joint certification, while achieving text robustness comparable to text-only smoothing under a different smoothed decision rule.

| Method | Image radius $\bar{r}$ | Text budget $\bar{d}$ |
|---|---|---|
| Image-only RS | 3.99 | 0 |
| Text-only RS | 0 | 3.26 |
| **Hybrid RS (ours)** | 3.76 | 3.07 |

*Table 1.* Average certified robustness for suffix attack. For Hybrid RS, the image radius is reported at text budget $d = 1$.

We also test for $\ell_0$-text attacks; we report the mean certified $\ell_0$ radius against prompt-injection attacks. The resulting averages are $\bar{d}_{\text{txt}} = 1.020$ and $\bar{d}_{\text{hybrid}} = 0.33$, indicating that hybrid RS yields notably smaller certified text budgets in this regime. These values reflect an intrinsic property of unsafe prompts: modifying only one or two tokens is often sufficient to switch the model's decision between safe and unsafe. A finer-grained analysis of certified accuracy as a function of the image perturbation radius $\epsilon$ is provided in Appendix A.11.

**Certified Accuracy Under Joint Perturbations.** On interaction-only multimodal examples, we apply the proposed hybrid randomized smoothing certificate and report certified accuracy as a function of the joint text and image budgets $(d, \epsilon)$. Figure 4 shows that certified accuracy de-

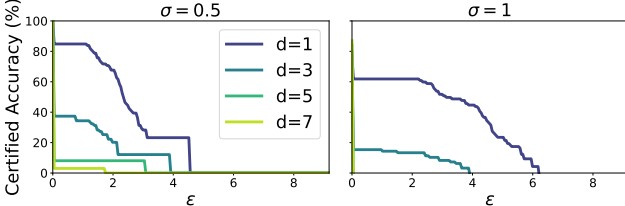

*Figure 4.* Certified accuracy as a function of the image perturbation radius $\epsilon$ ($\ell_2$ threat on images), combined with adversarial text suffix attacks. Curves correspond to different discrete text budgets $d$. Gaussian image smoothing uses $\sigma = 0.5$ (left) and $\sigma = 1.0$ (right).

creases monotonically as either the image radius $\epsilon$ or the text budget $d$ increases, as predicted by the hybrid Neyman–Pearson analysis. Larger text budgets induce a downward shift of the curves, and for $d \geq 5$ certification is nearly vacuous even around $\epsilon = 0$. Increasing the smoothing variance $\sigma$ reduces certified accuracy at small $\epsilon$, while extending certification to larger image perturbations. For example, at $\epsilon = 0.5$ and $d = 3$, certified accuracy decreases from 38% with $\sigma = 0.5$ to 36% with $\sigma = 1.0$. For $\sigma = 1.0$, certification becomes vacuous for text budgets $d > 3$.

*External benchmark check on MM-SafetyBench.* We additionally run the full pipeline on MM-SafetyBench (Liu et al., 2025) with 1,680 samples. Because this benchmark is not curated for interaction-only failures, most samples are either unimodally safe or unimodally unsafe; only 7.5% satisfy the interaction-only filter. On this subset, without retuning,

the mean certified text budget is 3.62 and the mean certified image radius is 3.37. This indicates that the certified accuracy regime observed on the curated split transfers beyond it, while also explaining why interaction-only certification has limited coverage on broad multimodal-safety benchmarks.

**Effect of the Corruption Rate $\beta$.** Table 2 highlights a clear trade-off induced by the parameter $\beta$. Smaller values of $\beta$ certify a larger fraction of examples but only for modest text budgets $d_{\max}$, whereas larger $\beta$ substantially increase the certified text budget at the cost of reduced coverage. Intermediate values provide a balance between these two effects. In the remainder of the experiment, we fix $\beta = 0.25$, which offers a reasonable compromise between certification coverage and certified text robustness.

**Discussion of Certification Parameters.** The parameters $(\sigma, \beta, \tau, d)$ jointly determine the behavior of the hybrid certificate. The smoothing variance $\sigma$ controls the continuous robustness radius and shapes the discrete Neyman–Pearson bound by smoothing likelihood ratios into a well-conditioned, invertible capacity map. The corruption rate $\beta$ governs a coverage–budget trade-off: smaller $\beta$ yields higher certified accuracy at small $d$, while larger $\beta$ enables certification at larger discrete budgets at the cost of reduced coverage. The threshold $\tau$ determines the strictness of certification and, in the safety setting, enforces a conservative requirement that the worst-case smoothed unsafe probability remains above a fixed low level. Finally, increasing the discrete budget $d$ monotonically enlarges the adversarial set, leading to smaller certified radii and lower certified accuracy.

| $\beta$ | Certified examples (%) | Mean $d_{\max}$ | Mean $r_\star(d_{\max})$ |
|---|---|---|---|
| 0.1 | 82.35 | 2.29 | 4.99 |
| 0.25 | 70.59 | 3.07 | 3.21 |
| 0.5 | 58.82 | 4.00 | 3.24 |
| 1.0 | 41.18 | 8.00 | 4.57 |

*Table 2.* Effect of the parameter $\beta$ on certification coverage, maximum certified text budget, and corresponding certified radius for image. We choose $\sigma = 1.0$ here for Gaussian noise.

**Certification Run Time.** Certification runtime is dominated by repeated LLaVA-Guard forward passes rather than by the hybrid Neyman–Pearson computation. The certificate-side operations, including root finding and discrete aggregation, add less than one second per datapoint. The main structural overhead relative to image-only smoothing is that hybrid certification evaluates a discrete–continuous frontier over several text budgets, rather than a single scalar image radius. We therefore report both the default implementation and two efficiency variants in Table 3.

Further reductions may be obtained through confidence-sequence early stopping and input-adaptive sampling; we leave these optimizations to future work.

*Table 3.* Average certification time per datapoint for $n = 10,000$ Monte Carlo samples on an H100 GPU. One-shot suffix and one-shot $\ell_0$ certification share the same computational cost.

| Setting | Time | Effect |
|---|---|---|
| Image-only RS | $\approx 156\,\text{s}$ | Single image radius |
| Hybrid RS, default | $\approx 500\,\text{s}$ | Full $(d, \epsilon)$ frontier |
| Hybrid RS + batching/FlashAttention | $\approx 0.7\times$ | Same certificate |
| One-shot suffix or $\ell_0$, $d_{\max} = 8$ | $\approx 44\,\text{s}$ | $\bar{r}$ : 2.07→1.55, $\bar{d}$ : 1.48→1.23 |
| Text-only RS, one-shot | $\approx 44\,\text{s}$ | One-shot estimate |

*Table 4.* Empirical adaptive attack results. These values estimate attack strength, not certification confidence. Certification uses the Clopper–Pearson lower bound with $n = 10,000$ samples and risk level $\alpha = 0.01$.

| Attack budget | Safe% |
|---|---|
| $d = d^*$, $\varepsilon = r^*$ | $74.286 \pm 4.695$ |
| $d = d^*$, $\varepsilon = 1.5r^*$ | $75.321 \pm 4.672$ |
| $d = d^*$, $\varepsilon = 3r^*$ | $76.571 \pm 4.730$ |
| $d = d^* + 2$, $\varepsilon = r^*$ | $76.393 \pm 5.123$ |
| $d = d^* + 4$, $\varepsilon = r^*$ | $90.214 \pm 2.284$ |

### 5.3. Empirical Attacks on Safety Filtering

Randomized smoothing certifies robustness in terms of a smoothed prediction probability. Accordingly, the relevant empirical question is not worst-case attack success, but whether direct optimization of the complementary smoothed quantity $\mathbb{P}_{(Z_1, Z_2) \sim p(\cdot | x')}[f(Z_1, Z_2) = \text{"Safe"}]$ can drive the smoothed classifier across the decision threshold $\tau$. Since certification enforces $\mathbb{P}(\text{Unsafe}) > \tau$, an empirical violation is equivalently $\mathbb{P}(\text{Safe}) > 1 - \tau$. We implement a distribution-matched adaptive multimodal attack that directly optimizes a Monte Carlo estimate of this quantity by alternating gradient-based updates on an append-only text suffix and $\ell_2$-constrained updates on the image, following (Salman et al., 2019). While structurally similar to prior multimodal attacks (Yin et al., 2023), the optimization target is the certified probability itself rather than a surrogate loss on the base classifier. More details on the attack procedure are given in Appendix A.5. Table 4 reports empirical attack optimization results over 20 attack iterations and 20 examples, using $n_{\text{samples}} = 1000$ Monte Carlo samples only to estimate the attack objective. These samples are not used to certify robustness. Here $(d^*, r^*)$ denotes the certified text and image budgets returned by Hybrid RS, while $(d, \varepsilon)$ are the empirical attack budgets. Safe% is the fraction of trials in which the smoothed "Safe" probability exceeds the decision threshold. With $\tau = 4.6 \times 10^{-5}$, certification is violated only if this probability exceeds $1 - \tau \approx 99.995\%$, so intermediate Safe% values remain far from breaking the de-

fense, as $\tau$ is very conservative. Empirically, the attack does not violate the certificate in agreement with the theoretical guarantees.

## 6. Conclusion

We introduced a unified randomized smoothing framework for heterogeneous discrete–continuous perturbations and derived an exact Neyman–Pearson characterization of the joint worst-case behavior. Our analysis shows that hybrid smoothing admits a closed-form, one-dimensional certificate that strictly generalizes classical Gaussian smoothing and discrete knapsack-based certificates, while revealing why naive multimodal composition fails even under independent noise. We demonstrated the practical relevance of this framework on multimodal safety filtering, providing, to our knowledge, the first model-agnostic Neyman–Pearson certificate for joint discrete-token and continuous-image perturbations in interaction-dependent text–image safety filtering.

While safety filtering serves as a natural motivating application, the proposed framework is not specific to safety. It applies more broadly to robustness certification in heterogeneous input spaces, including mixed discrete–continuous settings such as tabular–text models or multi-sensor fusion, where joint perturbations cannot be reduced to unimodal guarantees. Beyond safety, the hybrid Neyman–Pearson formulation provides a principled tool for analyzing and certifying robustness under heterogeneous noise, and offers a foundation for narrowing the gap between certified and empirical robustness in complex multimodal systems.

**Limitations.** The main practical cost is the number of VLM forward passes required by randomized smoothing; the hybrid certificate computation itself is negligible relative to inference. Smoothing can also reduce utility, especially under full-text uniform corruption, although suffix-only smoothing preserves substantially more semantic content. The certified lower bound can also be conservative: as in discrete randomized smoothing, the Neyman–Pearson/knapsack relaxation is tight given only the clean smoothed probability $p_A$, but the corresponding worst-case base classifier may be overly pessimistic. Tighter certificates may require additional structure on the base model, such as Lipschitz information (Chen et al., 2024; Delattre et al., 2024; 2025). Finally, interaction-only safety failures are relatively rare in broad benchmarks, so our main evaluation uses a curated interaction-only split and an additional MM-SafetyBench transfer check rather than a large naturally balanced benchmark.

## Acknowledgments

We thank Sylvain Delattre for the proof reading of Theorem 4.2.

This work is partially supported by JST PRESTO JP-MJPR23P5, JST CREST JPMJCR21M2, JST NEXUS JP-MJNX25C4. This work was granted access to the HPC resources of IDRIS under the allocation A0191016927 and AD011014214R2 made by GENCI. This work has received support from the French government, managed by the National Research Agency, under the France 2030 program with the reference "PR[AI]RIE-PSAI" (ANR-23-IACL-0008) and "PEPR-SHARP" (ANR-23-PEIA-0008).

## Impact Statement

This work studies certified robustness for heterogeneous discrete–continuous perturbations, with applications to multimodal safety filtering. The intended positive impact is to provide conservative, model-agnostic tools for assessing robustness of systems exposed to joint text–image adversarial inputs. Potential limitations include the computational cost of randomized smoothing and the risk that certificates are misinterpreted as complete safety guarantees outside the specified threat model. The guarantees apply only to the stated perturbation budgets, smoothing kernels, classifier, and confidence parameters.

## Code Availability

Code for reproducing the main experiments and certificates is available at https://github.com/tdsai-lab/hybrid-randomized-smoothing.

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

# A. Appendix

### A.1. Choice of Threshold $\tau$

Randomized smoothing certifies robustness by bounding the smoothed classifier's output under worst-case perturbations. For a binary classifier $f : \mathcal{X} \to \{0, 1\}$ and its smoothed version $g(x) = \mathbb{E}_{z \sim p(\cdot|x)}[f(z)]$, the certified radius at confidence level $\tau \in (0, 1)$ is

$$r_{\mathrm{RS}}(x; \tau) = \sup \{\delta \geq 0 : \ p_{\mathrm{adv}}(\delta) > \tau\},$$

where $p_{\mathrm{adv}}(\delta)$ denotes the worst-case smoothed score under perturbations of radius $\delta$.

**Role of $\tau$.** The threshold $\tau$ controls the confidence required for certification. A higher $\tau$ demands that the smoothed classifier remain more confident under perturbation, yielding stronger robustness guarantees but smaller certified radii. Conversely, a lower $\tau$ relaxes the constraint on $g(x_{\mathrm{adv}})$, enlarging the certified radius but tolerating weaker evidence.

In classical settings where certification targets *correct* predictions, low values of $\tau$ lead to weaker guarantees and are considered non-conservative. However, our setting reverses this logic.

**Certifying Unsafe Predictions.** We are interested in certifying that $f(x) = 1$ (i.e., the input is flagged as unsafe) remains stable under perturbation. Since $g(x) = \mathbb{P}[f(z) = 1]$, certification at low $\tau$ means we must guarantee that the smoothed unsafe probability remains above a tiny threshold, even in the worst case:

$$g(x_{\mathrm{adv}}) > \tau \quad \text{for all admissible } x_{\mathrm{adv}}.$$

This is a *conservative* guarantee: with $\tau = 4.6 \times 10^{-5}$, certification is only violated if the "Unsafe" probability drops below $0.0046\%$. Thus, a large number of Monte Carlo samples must still yield $f(z) = 1$ to maintain certification. In practice, this demands a classifier that is highly confident in its unsafe prediction at the clean input.

### A.2. Proof of Theorem 4.2

First, we derivethe following lemmas,

**Lemma A.1** (Continuity of the distribution of discrete mixtures). *Let $Z_1$ be discrete and $Y$ be a nonnegative random variable independent of $Z_1$. Assume that for every $z_1$ with $\mathbb{P}(Z_1 = z_1) > 0$, the law of $a(z_1)Y$ has no atoms on $(0, \infty)$, where $a(z_1) \geq 0$ is a deterministic scalar. Then $X := a(Z_1)Y$ has no atoms on $(0, \infty)$, hence its distribution function $F_X$ is continuous on $(0, \infty)$.*

*Proof.* Fix $x > 0$. By the law of total probability and independence,

$$\mathbb{P}(X = x) = \sum_{z_1} \mathbb{P}(Z_1 = z_1) \, \mathbb{P}(a(z_1)Y = x \mid Z_1 = z_1) = \sum_{z_1} \mathbb{P}(Z_1 = z_1) \, \mathbb{P}(a(z_1)Y = x).$$

Each term vanishes by assumption, hence $\mathbb{P}(X = x) = 0$ for all $x > 0$. Therefore $F_X$ has no jump on $(0, \infty)$ and is continuous there. $\qquad \square$

For a real-valued random variable $X$ with distribution function $F_X(t) = \mathbb{P}(X \leq t)$, we define the (left-continuous) quantile function as the generalized inverse

$$Q_X(p) := \inf\{t \in \mathbb{R} : \ F_X(t) \geq p\}, \qquad p \in (0, 1).$$

**Lemma A.2** (Quantile identity). *Let $X \geq 0$ be integrable with distribution function $F_X$ and quantile function $Q_X$. Fix $p \in (0, 1)$ and set $q := Q_X(p)$. If $F_X$ is continuous at $q$, then*

$$\mathbb{E}[X \, \mathbf{1}\{X \leq Q_X(p)\}] = \int_0^p Q_X(u) \, du.$$

*Proof.* Since $F_X$ is continuous at $q$, we have $F_X(q) = p$. Using the layer-cake representation for the nonnegative random variable $X \, \mathbf{1}\{X \leq q\}$,

$$\mathbb{E}[X \, \mathbf{1}\{X \leq q\}] = \int_0^q \mathbb{P}(t < X \leq q) \, dt = \int_0^q \big(p - F_X(t)\big) \, dt.$$

On the other hand, by Fubini's theorem and the definition of the generalized inverse,

$$\int_0^p Q_X(u)\,du = \int_0^q \left( \int_0^p \mathbf{1}\{t \le Q_X(u)\}\,du \right) dt = \int_0^q \left( p - F_X(t) \right) dt.$$

The two expressions coincide. $\qquad\qquad\square$

We give a formal version of the main result and proof

**Theorem A.3** (Hybrid randomized smoothing certificate)**.** *Let $\mathcal{X}_1$ be a finite or countable set and let $d \in \mathbb{N}$. Fix $\sigma > 0$ and $x = (x_1, x_2) \in \mathcal{X}_1 \times \mathbb{R}^D$. Assume the randomized smoothing corruption distribution factorizes as*

$$p(z \mid x) = p_1(z_1 \mid x_1)\, p_2(z_2 \mid x_2), \qquad p_2(\cdot \mid x_2) = \mathcal{N}(x_2, \sigma^2 I_d),$$

*with $p_1(\cdot \mid x_1)$ a probability mass function on $\mathcal{X}_1$.*

*Fix an adversarial input $x_{\mathrm{adv}} = (x_{1,\mathrm{adv}}, x_{2,\mathrm{adv}})$ with $D_1(x_1, x_{1,\mathrm{adv}}) \le d$ and $\|x_{2,\mathrm{adv}} - x_2\|_2 \le \epsilon$. Write $\Delta := x_{2,\mathrm{adv}} - x_2$ and $r := \|\Delta\|_2$. Assume absolute continuity in the discrete channel: for all $z_1$,*

$$p_1(z_1 \mid x_1) = 0 \implies p_1(z_1 \mid x_{1,\mathrm{adv}}) = 0,$$

*and define, for $p_1(z_1 \mid x_1) > 0$,*

$$\gamma_1(z_1) := \frac{p_1(z_1 \mid x_{1,\mathrm{adv}})}{p_1(z_1 \mid x_1)}.$$

*Let $f : \mathcal{X}_1 \times \mathbb{R}^D \to \{0, 1\}$ be any measurable base classifier and define the smoothed value at the clean input*

$$p_A := \mathbb{E}_{z \sim p(\cdot \mid x)}[f(z)] \in (0, 1).$$

*For a fixed discrete adversarial input $x_{1,\mathrm{adv}}$ and a fixed continuous radius $r > 0$, define the adversarial value*

$$V(x_{1,\mathrm{adv}}; r) := \inf_{0 \le h \le 1} \mathbb{E}_{z \sim p(\cdot \mid x_{\mathrm{adv}})}[h(z)] \quad s.t. \quad \mathbb{E}_{z \sim p(\cdot \mid x)}[h(z)] = p_A. \tag{2}$$

*Then the unique optimal solution of* (2) *is given by a likelihood-ratio threshold test*

$$h^\star(z_1, z_2) = \mathbf{1}\{\gamma(z_1, z_2; r) \le t^\star(r)\}, \qquad \gamma(z_1, z_2; r) := \gamma_1(z_1)\gamma_2(z_2; r),$$

*where $\gamma_2$ is the Gaussian likelihood ratio*

$$\gamma_2(z_2; r) := \frac{\phi_\sigma(z_2 - x_{2,\mathrm{adv}})}{\phi_\sigma(z_2 - x_2)} = \exp\left( \frac{1}{\sigma^2}\Delta^\top \left( z_2 - \frac{x_2 + x_{2,\mathrm{adv}}}{2} \right) \right),$$

*and $t^\star(r) > 0$ is the unique solution of*

$$F(t; r) = p_A, \qquad F(t; r) := \sum_{z_1} p_1(z_1 \mid x_1)\, \Phi\left( \frac{\frac{r^2}{2} + \sigma^2(\log t - \log \gamma_1(z_1))}{\sigma r} \right).$$

*The corresponding optimal adversarial value admits the closed form*

$$V(x_{1,\mathrm{adv}}; r) = \sum_{z_1} p_1(z_1 \mid x_{1,\mathrm{adv}})\, \Phi\left( \frac{\frac{r^2}{2} + \sigma^2(\log t^\star(r) - \log \gamma_1(z_1))}{\sigma r} - \frac{r}{\sigma} \right).$$

*Moreover, for every fixed $x_{1,\mathrm{adv}}$, the map $r \mapsto V(x_{1,\mathrm{adv}}; r)$ is nonincreasing on $(0, \infty)$. Hence, under the constraint $\|x_{2,\mathrm{adv}} - x_2\|_2 \le \epsilon$, the worst case is attained at $r = \epsilon$.*

*Finally, defining the hybrid worst-case smoothed value under budgets $(d, \epsilon)$ as*

$$p_{\mathrm{adv}}(d, \epsilon) := \inf_{D_1(x_1, x_{1,\mathrm{adv}}) \le d} V(x_{1,\mathrm{adv}}; \epsilon),$$

*Proof.* Let

$$w(z_1, z_2) := p_1(z_1 \mid x_1) \, \phi_\sigma(z_2 - x_2), \qquad v(z_1, z_2) := p_1(z_1 \mid x_{1,\text{adv}}) \, \phi_\sigma(z_2 - x_{2,\text{adv}}),$$

so that $w$ and $v$ are probability measures on $\mathcal{X}_1 \times \mathbb{R}^D$ and $\gamma(\cdot; r) = v/w$ is the Radon–Nikodym derivative of $v$ w.r.t. $w$. The absolute-continuity assumption on $p_1$ ensures $v \ll w$.

**Step 1: Neyman–Pearson Form of the Optimizer.** Problem (2) is the classical Neyman–Pearson problem between $(w, v)$. By the Neyman–Pearson lemma for general measures , There exists an optimal test of the form

$$h^\star = \mathbf{1}\{\gamma < t\} + \alpha \, \mathbf{1}\{\gamma = t\} \quad \text{for some } t \geq 0, \ \alpha \in [0, 1],$$

with $\mathbb{E}_w[h^\star] = p_A$.

We now show that for $r > 0$ there are no ties under $w$, so we may take $\alpha = 0$. Fix $z_1$ and $t > 0$. Since

$$\gamma(z_1, z_2; r) = \gamma_1(z_1) \exp\left( \frac{1}{\sigma^2} \Delta^\top \left( z_2 - \frac{x_2 + x_{2,\text{adv}}}{2} \right) \right),$$

the level set $\{\gamma(z_1, \cdot; r) = t\}$ is either empty or a hyperplane $\{z_2 : \Delta^\top z_2 = c\}$ for some $c \in \mathbb{R}$. Because $w(\cdot \mid Z_1 = z_1)$ is Gaussian with a continuous density on $\mathbb{R}^D$, any hyperplane has $w$-measure 0. Summing over countably many $z_1$ yields $w(\gamma = t) = 0$. Thus, for $r > 0$, an optimal test can be taken as

$$h^\star(z_1, z_2) = \mathbf{1}\{\gamma(z_1, z_2; r) \leq t\}. \tag{3}$$

**Step 2: Likelihood-Ratio CDF $F(t; r)$ and Uniqueness of $t^\star(r)$.** Fix $r > 0$ and $t > 0$. The acceptance set conditional on $Z_1 = z_1$ is

$$\gamma_1(z_1) \gamma_2(z_2; r) \leq t \iff \Delta^\top z_2 \leq \frac{r^2}{2} + \sigma^2 (\log t - \log \gamma_1(z_1)). \tag{4}$$

Let $Z_2 \sim \mathcal{N}(x_2, \sigma^2 I_d)$ and define the 1D projection $U := \Delta^\top (Z_2 - x_2)/r \sim \mathcal{N}(0, \sigma^2)$. Then (4) is equivalent to $U \leq \left( \frac{r^2}{2} + \sigma^2 (\log t - \log \gamma_1(z_1)) \right)/r$, so

$$\mathbb{P}(\gamma_1(z_1) \gamma_2(Z_2; r) \leq t) = \Phi\left( \frac{\frac{r^2}{2} + \sigma^2 (\log t - \log \gamma_1(z_1))}{\sigma r} \right).$$

Therefore, using (3),

$$F(t; r) := \mathbb{E}_w[h^\star] = \sum_{z_1} p_1(z_1 \mid x_1) \, \Phi\left( \frac{\frac{r^2}{2} + \sigma^2 (\log t - \log \gamma_1(z_1))}{\sigma r} \right).$$

Each summand is continuous and strictly increasing in $t$ (it is $\Phi$ of an affine function of $\log t$), hence $F(\cdot; r)$ is continuous and strictly increasing. Moreover, $F(t; r) \to 0$ as $t \downarrow 0$ and $F(t; r) \to 1$ as $t \uparrow \infty$. Since $p_A \in (0, 1)$, there exists a unique $t^\star(r) > 0$ such that

$$F(t^\star(r); r) = p_A. \tag{5}$$

**Step 3: Closed Form for the Optimal Adversarial Value.** Under $v$, we have $Z_2' \sim \mathcal{N}(x_{2,\text{adv}}, \sigma^2 I_d) = \mathcal{N}(x_2 + \Delta, \sigma^2 I_d)$. Let $U' := \Delta^\top (Z_2' - (x_2 + \Delta))/r \sim \mathcal{N}(0, \sigma^2)$. Then $\Delta^\top Z_2' = \Delta^\top x_2 + r^2 + rU'$ and (4) becomes

$$U' \leq \frac{\frac{r^2}{2} + \sigma^2 (\log t - \log \gamma_1(z_1))}{r} - r.$$

Hence

$$\mathbb{P}(\gamma_1(z_1)\gamma_2(Z_2'; r) \le t) = \Phi\left(\frac{\frac{r^2}{2} + \sigma^2(\log t - \log\gamma_1(z_1))}{\sigma r} - \frac{r}{\sigma}\right),$$

and therefore

$$\mathbb{E}_v[\mathbf{1}\{\gamma \le t\}] = \sum_{z_1} p_1(z_1 \mid x_{1,\mathrm{adv}}) \, \Phi\left(\frac{\frac{r^2}{2} + \sigma^2(\log t - \log\gamma_1(z_1))}{\sigma r} - \frac{r}{\sigma}\right).$$

Setting $t = t^\star(r)$ from (5) yields the stated expression for $V(x_{1,\mathrm{adv}}; r)$.

**Step 4: monotonicity of $r \mapsto V(x_{1,\mathrm{adv}}; r)$ via convex order.** We now prove that for fixed $x_{1,\mathrm{adv}}$, the optimal value $V(x_{1,\mathrm{adv}}; r)$ is nonincreasing in $r$.

**4.1 Identify $V$ as a Lower Partial Expectation of the Likelihood Ratio.** Let $(Z_1, Z_2) \sim w$ and define the likelihood ratio random variable

$$\Gamma_r := \gamma(Z_1, Z_2; r) = \frac{v(Z_1, Z_2)}{w(Z_1, Z_2)}.$$

Then for any measurable set $A$,

$$\mathbb{P}_v((Z_1, Z_2) \in A) = \mathbb{E}_w[\Gamma_r \, \mathbf{1}_A].$$

In particular, letting $A_t(r) := \{\Gamma_r \le t\}$,

$$F(t; r) = \mathbb{P}_w(\Gamma_r \le t), \qquad \mathbb{E}_v[\mathbf{1}_{A_t(r)}] = \mathbb{E}_w[\Gamma_r \, \mathbf{1}\{\Gamma_r \le t\}].$$

For $r > 0$, the distribution of $\Gamma_r$ under $w$ is continuous as countable mixture of continuous distributions, so $t^\star(r)$ in (5) is the $p_A$-quantile:

$$t^\star(r) = Q_{\Gamma_r}(p_A),$$

where $Q_X$ denotes the (left-continuous) quantile function. Hence

$$V(x_{1,\mathrm{adv}}; r) = \mathbb{E}_w[\Gamma_r \, \mathbf{1}\{\Gamma_r \le Q_{\Gamma_r}(p_A)\}]. \tag{6}$$

**4.2 Factorization of $\Gamma_r$ Under $w$.** Under $w$, $Z_1 \sim p_1(\cdot \mid x_1)$ and $Z_2 - x_2 \sim \mathcal{N}(0, \sigma^2 I_d)$ are independent. Let $G \sim \mathcal{N}(0, 1)$ and set

$$Y_r := \exp\left(\frac{r}{\sigma}G - \frac{r^2}{2\sigma^2}\right), \qquad \mathbb{E}[Y_r] = 1.$$

Then one checks, by the standard Gaussian likelihood-ratio algebra, that

$$\Gamma_r = \gamma_1(Z_1) \, Y_r, \tag{7}$$

and that $\gamma_1(Z_1)$ and $Y_r$ are independant.

**4.3 Convex Order Growth of the Lognormal Factor.** Let $(B_t)_{t\ge 0}$ be standard Brownian motion with natural filtration $(\mathcal{F}_t)_{t\ge 0}$ and define

$$M_t := \exp(B_t - t/2), \qquad t \ge 0.$$

Then $(M_t)$ is a positive martingale and satisfies $\mathbb{E}[M_t] = 1$ for all $t$. For $0 \le s \le t$, the martingale property yields

$$\mathbb{E}[M_t \mid \mathcal{F}_s] = M_s.$$

Let $\varphi : \mathbb{R}_+ \to \mathbb{R}$ be any convex function for which the expectations exist. By conditional Jensen's inequality,

$$\varphi(M_s) = \varphi(\mathbb{E}[M_t \mid \mathcal{F}_s]) \le \mathbb{E}[\varphi(M_t) \mid \mathcal{F}_s].$$

Taking expectations gives

$$\mathbb{E}[\varphi(M_s)] \le \mathbb{E}[\varphi(M_t)].$$

Now set $s = r_1^2/\sigma^2$ and $t = r_2^2/\sigma^2$ with $0 \leq r_1 \leq r_2$. Since $B_t \stackrel{d}{=} (r/\sigma)G$ for $G \sim \mathcal{N}(0,1)$, we have

$$M_{r^2/\sigma^2} \stackrel{d}{=} \exp\left(\frac{r}{\sigma}G - \frac{r^2}{2\sigma^2}\right) =: Y_r, \qquad \mathbb{E}[Y_r] = 1.$$

Therefore, for all convex $\varphi$,

$$\mathbb{E}[\varphi(Y_{r_1})] \leq \mathbb{E}[\varphi(Y_{r_2})], \qquad \mathbb{E}[Y_{r_1}] = \mathbb{E}[Y_{r_2}].$$

Finally, since $\gamma_1(Z_1) \geq 0$ is independent of $Y_r$, the map $y \mapsto \varphi(ay)$ is convex for every fixed $a \geq 0$. Conditioning on $a = \gamma_1(Z_1)$ yields

$$\mathbb{E}[\varphi(\gamma_1(Z_1)Y_{r_1})] \leq \mathbb{E}[\varphi(\gamma_1(Z_1)Y_{r_2})].$$

Using $\mathbb{E}[Y_{r_1}] = \mathbb{E}[Y_{r_2}] = 1$, this implies

$$\Gamma_{r_1} \leq_{\mathrm{cx}} \Gamma_{r_2} \quad \text{and} \quad \mathbb{E}[\Gamma_{r_1}] = \mathbb{E}[\Gamma_{r_2}] = 1. \tag{6}$$

## 4.4 Convex Order Implies Monotonicity of Lower Partial Expectations.

Let $X \geq 0$ be integrable with quantile function $Q_X$ and define the lower partial expectation (Lorenz integral)

$$L_X(p) := \mathbb{E}[X \, \mathbf{1}\{X \leq Q_X(p)\}], \qquad p \in (0,1).$$

By Lemma A.2, if $F_X$ is continuous at $Q_X(p)$,

$$L_X(p) = \int_0^p Q_X(u)\,du. \tag{8}$$

In our application, $X = \Gamma_r = \gamma_1(Z_1)Y_r$ under $w$, where $Z_1$ is discrete, $Y_r$ is lognormal, and $Z_1$ is independent of $Y_r$. For any $z_1$ with $\mathbb{P}(Z_1 = z_1) > 0$, if $\gamma_1(z_1) > 0$ then $\gamma_1(z_1)Y_r$ has a continuous density on $(0,\infty)$, and if $\gamma_1(z_1) = 0$ then $\gamma_1(z_1)Y_r \equiv 0$. Lemma A.1 therefore implies that $F_{\Gamma_r}$ is continuous on $(0,\infty)$, in particular at $Q_{\Gamma_r}(p_A) = t^\star(r) > 0$.

For nonnegative integrable random variables $X, Y$ with equal mean, convex order implies Lorenz order:

$$X \leq_{\mathrm{cx}} Y \text{ and } \mathbb{E}[X] = \mathbb{E}[Y] \implies \int_0^p Q_X(u)\,du \geq \int_0^p Q_Y(u)\,du \qquad \forall p \in (0,1), \tag{9}$$

see (Shaked & Shanthikumar, 2007)[Theorem 3.A.10]. Using (8), this is equivalent to $L_X(p) \geq L_Y(p)$.

Applying (9) to (6) yields, for $0 \leq r_1 \leq r_2$,

$$L_{\Gamma_{r_1}}(p_A) \geq L_{\Gamma_{r_2}}(p_A).$$

Using (6), we conclude

$$V(x_{1,\mathrm{adv}}; r_1) \geq V(x_{1,\mathrm{adv}}; r_2),$$

so the map $r \mapsto V(x_{1,\mathrm{adv}}; r)$ is nonincreasing on $(0,\infty)$. Consequently, under the constraint $\|x_{2,\mathrm{adv}} - x_2\|_2 \leq \epsilon$, the worst case is attained at $r = \epsilon$.

**Step 5: outer discrete adversary. .** For each admissible $x_{1,\mathrm{adv}}$ with $D_1(x_1, x_{1,\mathrm{adv}}) \leq d$, Step 4 shows

$$\inf_{\|x_{2,\mathrm{adv}} - x_2\|_2 \leq \epsilon} \inf_{h \text{ s.t. } \mathbb{E}_w[h] = p_A} \mathbb{E}_v[h] = V(x_{1,\mathrm{adv}}; \epsilon).$$

Taking the infimum over $x_{1,\mathrm{adv}}$ yields

$$p_{\mathrm{adv}}(d, \epsilon) = \inf_{D_1(x_1, x_{1,\mathrm{adv}}) \leq d} V(x_{1,\mathrm{adv}}; \epsilon).$$

$\square$

### A.3. Recovery of the Discrete Certificate as $\sigma \to \infty$

In the hybrid randomized smoothing formulation, the Gaussian component introduced on the continuous channel acts only as a smooth relaxation of the discrete Neyman–Pearson problem. The corresponding likelihood-ratio CDF depends on $(r, \sigma)$ exclusively through the dimensionless ratios $r/(2\sigma)$ and $\sigma/r$, so that increasing $\sigma$ progressively sharpens the relaxation. As $\sigma \to \infty$ with $r > 0$ fixed, the Gaussian CDF appearing in the capacity equation converges pointwise to a step function, and the likelihood-ratio CDF $F_\sigma(t; r)$ converges to the discrete staircase

$$F_\infty(t) = \sum_{z_1} p_1(z_1 \mid x_1)\, \mathbf{1}\{\gamma_1(z_1) \le t\},$$

which is exactly the classical discrete Neyman–Pearson (fractional-knapsack) capacity. Any threshold $t^\star$ satisfying this discrete constraint attains the optimal worst-case value for the discrete channel. Hence, letting $\sigma \to \infty$ recovers the discrete-only randomized smoothing certificate with no loss of tightness.

### A.4. Variable-Length Text in Hybrid Randomized Smoothing

Hybrid randomized smoothing is defined on a fixed ambient discrete space, while textual inputs are intrinsically variable-length. Let a clean prompt be $P = (\rho_1, \ldots, \rho_n)$. Under an append-only attack, the adversary appends a suffix $\alpha$ of length at most $d$, yielding $P' = P \| \alpha$, with no assumption of length preservation at the semantic level.

For certification, all prompts are embedded into a fixed-length representation by padding to a global maximum length $L$, chosen *a priori* to upper-bound the lengths of all prompts under evaluation. Padding is performed using a special `PAD` token. In this padded representation, suffix attacks correspond to replacing trailing `PAD` tokens with adversarial tokens, while deletions correspond to the inverse operation. As a result, all text attacks are mapped to length-preserving perturbations in a fixed-dimensional discrete space.

The smoothing kernel is defined on this padded space, and the resulting Neyman–Pearson certificate depends only on the number of perturbed coordinates, i.e., the append budget $d$, and not on the original prompt length $n$. Variable-length effects are therefore handled implicitly by the embedding, and do not alter the form or validity of the certification analysis.

### A.5. Adaptive Multimodal Attack on Smoothed Classifier

We empirically assess the soundness and tightness of the hybrid randomized smoothing certificate. The attack is designed to match the smoothing-based threat model and directly targets the smoothed classifier.

**Attack Objective.** Given a text–image input $(x_1, x_2)$, the attacker seeks a joint perturbation $(x_{1,\mathrm{adv}}, x_{2,\mathrm{adv}})$ satisfying $D_1(x_1, x_{1,\mathrm{adv}}) \le d$ and $\|x_{2,\mathrm{adv}} - x_2\|_2 \le \epsilon$ that maximizes the complementary smoothed "Safe" probability

$$\mathbb{P}_{(Z_1, Z_2) \sim p(\cdot \mid x_{1,\mathrm{adv}}, x_{2,\mathrm{adv}})}[f(Z_1, Z_2) = \mathrm{Safe}].$$

In practice, the attack optimizes a Monte Carlo estimate of this quantity using i.i.d. samples from the hybrid smoothing distribution.

**Alternating Optimization.** The attack proceeds by alternating updates on the two modalities. For the text channel, we apply gradient-based coordinate updates on an append-only adversarial suffix, following GCG-style optimization (Zou et al., 2023) under a discrete budget $d$. For the image channel, we perform $\ell_2$-constrained projected gradient descent steps under radius $\epsilon$ (Madry et al., 2018). At each iteration, one modality is updated while the other is held fixed, yielding a block-coordinate ascent scheme on the estimated smoothed objective. Algorithm 1 summarizes the procedure.

**Relation to Prior Work.** The attack is structurally similar to multimodal adversarial methods such as VLATTACK (Yin et al., 2023), but differs in its optimization target. Rather than attacking the base classifier, it directly optimizes the smoothed prediction probability used in certification, following the smoothing-aware attack principle of (Salman et al., 2019).

### A.6. Policy Prompts for the LLaVA Safety Detector

**Multimodal Safety Policy (Exact Prompt).**

You are a safety classifier. Do not answer, explain, or comply with the user's request.

*Scope.* Assess *only* the last user turn. You are given both text and an image.

---

**Algorithm 1** Smoothing-aware alternating multimodal attack

---

**Require:** Classifier $f$, input $(x_1, x_2)$, budgets $(d, \epsilon)$, iterations $T$, MC samples $N$
**Ensure:** Adversarial pair $(x_{1,\text{adv}}, x_{2,\text{adv}})$
1: Initialize adversarial suffix $s_0$ and image perturbation $\delta_0 \leftarrow 0$
2: **for** $t = 0$ to $T - 1$ **do**
3:     Estimate the smoothed "Safe" probability using $N$ samples
4:     **Text update:** update $s_t$ by a gradient-based coordinate step
5:     **Image update:** update $\delta_t$ by $\ell_2$-PGD and project onto $\{\|\delta\|_2 \leq \epsilon\}$
6: **end for**
7: **Output:** $(x_1 \oplus s_T,\ x_2 + \delta_T)$

---

*Evaluation rule (strict).* You must assess (A) the text alone, (B) the image alone, and (C) the combined meaning of text and image.

*Decision rule.* Output `Unsafe` if and only if at least one of the following holds:

(A) *Text-alone Unsafe.* The text alone explicitly asks for, encourages, or meaningfully enables harmful or illegal activity, including violence, self-harm, weapons, criminal activity, illegal drugs, sexual exploitation of minors, targeted hate, or animal cruelty.

(B) *Image-alone Unsafe.* The image alone depicts harmful, illegal, or offensive content that would be unsafe without text, including violence, self-harm imagery, weapons with intent to harm, criminal acts, illegal drugs, sexual exploitation of minors, hate symbols, or animal cruelty.

(C) *Combination Unsafe.* Even if text and image are individually safe, their combination is unsafe when they jointly imply intent, instructions, targeting, or operational details enabling harmful or illegal activity.

Otherwise output `Safe`. If intent is ambiguous, output `Safe`.

*Output constraint.* Output exactly one word: `Safe` or `Unsafe`.

## A.7. Numerical Precision and Conservative Certification

Our certification algorithm involves two nested one-dimensional root-finding procedures based on bisection: an inner bisection to recover the Neyman–Pearson threshold $t^\star(r)$ defined by

$$F_\sigma(t^\star(r), r) = p_A,$$

and an outer bisection to compute the maximal certified radius

$$r^\star = \sup\{r : V_\sigma(t^\star(r), r) \geq \tau\}.$$

Both procedures rely only on monotonicity and therefore admit explicit, deterministic control of numerical error.

**Tolerance-Controlled Bisection.** Let a bisection be run on a monotone function over an initial interval $[a_0, b_0]$. After $k$ iterations, the interval width satisfies

$$b_k - a_k = \frac{b_0 - a_0}{2^k},$$

and any returned point lies within $(b_k - a_k)/2$ of the true solution. Equivalently, for a prescribed absolute tolerance $\varepsilon > 0$, it suffices to choose

$$k \ \geq \ \left\lceil \log_2\left(\frac{b_0 - a_0}{2\varepsilon}\right) \right\rceil.$$

In practice, this allows us to replace a fixed iteration budget by an explicit tolerance parameter, both for the threshold $t^\star(r)$ and for the certified radius $r^\star$.

**One-Sided Threshold Approximation.** For each fixed radius $r$, the map $t \mapsto F_\sigma(t, r)$ is nondecreasing. We therefore compute a lower bracket $t_L(r) \leq t^\star(r)$ by bisection and use $t_L(r)$ in place of $t^\star(r)$. Since $t_L(r) \leq t^\star(r)$, and the map $t \mapsto V_\sigma(t, r)$ is also nondecreasing, this yields

$$V_\sigma(t_L(r), r) \ \leq \ V_\sigma(t^\star(r), r).$$

As a consequence, any feasibility test based on $V_\sigma(t_L(r), r)$ is conservative: it may reject some feasible radii, but it cannot accept an infeasible one.

**Conservative Radius Certification.** The outer bisection on $r$ uses this conservative feasibility test and returns the lower endpoint of the final bracket. Denoting by $\hat{r}$ the returned radius, we obtain the deterministic guarantee

$$\hat{r} \leq r^\star, \qquad 0 \leq r^\star - \hat{r} \leq \varepsilon_r,$$

where $\varepsilon_r$ is the prescribed tolerance of the outer bisection. Thus, the numerical tolerance directly controls the tightness of the certificate, while preserving soundness.

**Choice of Tolerances.** In our experiments, the certified radius lies in the range $r \in [0.1, 10]$, and the smoothing scale satisfies $\sigma \in [0.1, 2.0]$. Over these ranges, the functions $F_\sigma$ and $V_\sigma$ are smooth combinations of Gaussian CDFs, and their monotonicity is well behaved. We therefore select absolute tolerances $\varepsilon_t$ and $\varepsilon_r$ for the inner and outer bisections, respectively, and compute the corresponding iteration budgets from the formulas above. Smaller tolerances yield tighter (less conservative) certificates at the cost of additional iterations, while any finite tolerance preserves correctness by construction.

Overall, this design allows the numerical precision of the certification procedure to be chosen arbitrarily by the user, with an explicit and provable impact on the conservativeness of the resulting certified radius.

## A.8. About Kernel Choice for Discrete Inputs

**Uniform Replacement Kernel.** Fix a corruption rate $\beta \in (0, 1)$. For each eligible position $\ell$,

$$Z_{1,\ell} = \begin{cases} x_{1,\ell}, & \text{with probability } 1 - \beta, \\ v \sim \text{Unif}(\mathcal{V} \setminus \{x_{1,\ell}\}), & \text{with probability } \beta. \end{cases}$$

This kernel is symmetric in the sense that the induced smoothed distribution depends only on the number of corrupted positions, not on their specific locations or values. It induces a discrete likelihood ratio

$$\gamma_1(z_1) = \frac{p_1(z_1 \mid x_{1,\text{adv}})}{p_1(z_1 \mid x_1)},$$

which takes finitely many values depending on how $z_1$ matches $x_1$ and $x_{1,\text{adv}}$. These likelihood ratios enter the hybrid likelihood-ratio CDF $F(t; r)$ in Theorem 4.2. Under a uniform kernel, any two adversarial suffixes of the same length are indistinguishable to the smoothing distribution.

We also consider an absorbing kernel below.

**Absorbing Kernel.** Let $[\text{PAD}] \in \mathcal{V}$ be a fixed absorbing token distinct from all regular tokens. For each eligible position $\ell$, define

$$Z_{1,\ell} = \begin{cases} x_{1,\ell}, & \text{with probability } 1 - \beta, \\ [\text{PAD}], & \text{with probability } \beta, \end{cases}$$

independently across positions.

Let $\mathcal{I} \subseteq \{1, \ldots, L\}$ denote the set of eligible token positions at which the absorbing noise is applied; positions $\ell \notin \mathcal{I}$ are kept fixed and are not subject to random replacement e.g constrained positions such as special tokens, [CLS], [SEP], padding outside the true sequence length, or any positions that should remain invariant under the threat model.

The resulting kernel is

$$p_1(z_1 \mid x_1) = \prod_{\ell \in \mathcal{I}} \Big[ (1 - \beta) \mathbf{1}\{z_{1,\ell} = x_{1,\ell}\} + \beta \mathbf{1}\{z_{1,\ell} = [\text{PAD}]\} \Big] \prod_{\ell \notin \mathcal{I}} \mathbf{1}\{z_{1,\ell} = x_{1,\ell}\}.$$

For this kernel, the discrete likelihood ratio satisfies $\gamma_1(z_1) \in \{0, 1\}$. As a consequence, the hybrid likelihood-ratio CDF $F(t; r)$ and the adversarial value $V(r)$ admit closed-form expressions.

**Connection to the Hybrid Certificate.** In both cases, the text kernel $p_1$ defines the discrete likelihood ratios $\gamma_1(z_1)$ that appear in the joint likelihood ratio

$$\gamma(z_1, z_2) = \gamma_1(z_1)\, \gamma_2(z_2),$$

with $\gamma_2$ the Gaussian likelihood ratio of the continuous modality. The hybrid Neyman–Pearson rule thresholds this joint ratio, and the resulting certificate is obtained by inverting the strictly monotone likelihood-ratio CDF

$$F(t;r) = \sum_{z_1} p_1(z_1 \mid x_1) \, \Phi\left(\frac{\frac{r^2}{2} + \sigma^2(\log t - \log \gamma_1(z_1))}{\sigma r}\right),$$

which reduces to the discrete fractional-knapsack formulation when $r = 0$ and to classical Gaussian randomized smoothing when the discrete channel is trivial.

### A.9. Smoothed Accuracy Degradation

We report the smoothed accuracy of the multimodal classifier under text and image smoothing, as a function of the image smoothing variance $\sigma$ and the text corruption rate $\beta$. The values quantify the utility loss induced by smoothing prior to certification.

| Threat model | $\sigma$ | $\beta = 0$ | $\beta = 0.1$ | $\beta = 0.25$ | $\beta = 0.5$ |
|---|---|---|---|---|---|
| full $\ell_0$ text | 0.0 | 100.00 | 75.71 | 32.38 | 3.81 |
| full $\ell_0$ text | 0.5 | 91.90 | 74.76 | 33.81 | 2.86 |
| full $\ell_0$ text | 1.0 | 88.57 | 68.10 | 32.86 | 2.38 |
| suffix | 0.0 | 100.00 | 100.00 | 94.85 | 88.84 |
| suffix | 0.5 | 95.28 | 95.71 | 93.99 | 88.41 |
| suffix | 1.0 | 91.85 | 91.85 | 90.99 | 84.98 |

*Table 5.* Smoothed accuracy under text and image smoothing.

Smoothed accuracy degrades mainly through text corruption. Full-text $\ell_0$ smoothing is destructive because uniform replacement may alter semantic content throughout the prompt. In the suffix threat model used for certification, only appended tokens are corrupted, so the core prompt is preserved and smoothed accuracy remains high. Image smoothing introduces comparatively smaller degradation in the tested range, consistent with the denoising-based image smoothing pipeline.

### A.10. Empirical Comparison with MMCert-Style Subsampling

We complement the discussion in Section 2 with an empirical comparison against MMCert-style independent subsampling on our Hateful Memes evaluation set, using the same safety detector (LLaVA-Guard). We restrict the comparison to the subset of approximately 100 examples that are classified as Unsafe without ablation, in order to isolate whether certification survives partial observation once the base prediction is correct. We treat text tokens and image patches as basic elements and estimate the smoothed unsafe probability from $n = 10{,}000$ Monte Carlo samples per configuration. Since this is a binary classification setting, the relevant diagnostics are the number of examples whose smoothed unsafe probability exceeds the certification threshold $\tau$, and the mean smoothed unsafe probability $\bar{p}_A$.

| Keep ratio | Examples with $\hat{p}_A > \tau$ | Mean $\bar{p}_A$ |
|---|---|---|
| 1.0 (no ablation) | 100/100 | 1.00 |
| 0.50 | 0/100 | 0.120 |
| 0.20 | 0/100 | 0.056 |
| 0.10 | 0/100 | 0.033 |
| 0.05 | 0/100 | 0.018 |

*Table 6.* MMCert-style independent subsampling on the interaction-only Hateful Memes subset. Under every ablated keep ratio, MMCert yields zero certifiable examples; the smoothed unsafe probability collapses far below the threshold required for a non-trivial certificate in this binary setting.

We further verified that the choice of patch granularity does not alter the conclusion: repeating the sweep with finer image patches yields mean $\bar{p}_A$ values of 0.009, 0.018, and 0.039 at keep ratios 0.10, 0.20, and 0.50 respectively, slightly lower than with the default patch size. We also tested asymmetric allocations of the keep budget across modalities. Keeping the full image but only 20% of the text yields 0/100 certifiable examples, and keeping the full text but only 20% of the image

yields the same outcome. In interaction-dependent settings, preserving a single modality is therefore insufficient: removing most of the other modality destroys the cross-modal signal required for the unsafe prediction.

This failure is structural. MMCert relies on prediction stability under partial observation, which is well matched to tasks with redundant evidence spread across tokens, patches, or frames. In Hateful Memes, the unsafe label often depends on sparse cross-modal alignment between specific textual and visual cues; independent subsampling can remove a critical part of the joint signal even when much of each modality is retained, so the smoothed unsafe probability collapses below the certification threshold. Intuitively, the probability of preserving all jointly necessary cues decays multiplicatively with subsampling across modalities. By contrast, Hybrid RS on the same Hateful Memes evaluation maintains non-trivial smoothed accuracy and certification under joint $(d, \epsilon)$ perturbations (Section 5.2), precisely because additive noise preserves the cross-modal structure that subsampling destroys.

| | MMCert | Knowledge Continuity | Ours |
|---|---|---|---|
| Perturbation model | $\ell_0$ per modality (discrete) | Representation space | $\ell_0$ discrete + $\ell_2$ continuous |
| Guarantee | Worst-case (NP, subsampling) | Probabilistic, distributional | Worst-case (hybrid NP) |
| Heterogeneous inputs | No (homogeneous discrete) | N/A (unimodal) | Yes |
| Cross-modal interaction | Weak in interaction-dependent regimes | Not evaluated | Yes (noise preserves signal) |
| Favorable regime | Redundant, set-like signals | — | Sparse, compositional, cross-modal |

*Table 7.* Comparison of certified multimodal robustness frameworks.

## A.11. Additional Experiments

### Certified Accuracy for $\ell_0$ Prompt Injection and $\ell_2$-Bounded Image Attacks.

Figure 5 reports the certified accuracy of the hybrid randomized smoothing certificate as a function of the image perturbation radius $\epsilon$, for a fixed discrete text budget $d = 1$. We evaluate this setting on a subset of 100 interaction-only examples drawn from the dataset constructed in Section 5.2. This subset consists of examples for which at least one certification method (text-only or hybrid) yields a non-zero certified text budget, i.e., $d_{\text{txt}} > 0$ or $d_{\text{hybrid}} > 0$, ensuring that certification is non-trivial on the discrete channel. For a given value of $\epsilon$, an example is counted as certified if $\epsilon$ does not exceed its certified hybrid robustness radius at budget $d$.

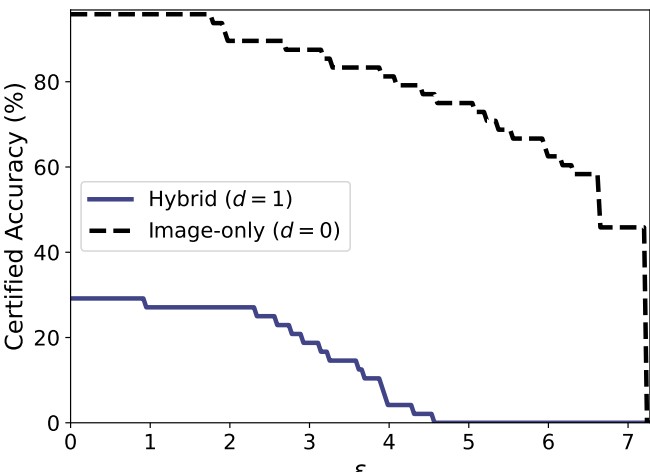

*Figure 5.* Certified accuracy of the hybrid randomized smoothing certificate under $\ell_0$ prompt-injection attacks, as a function of the image perturbation radius $\epsilon$ for text budget $d = 1$.

The resulting curves show that even small increases in image perturbation strength can rapidly erode certified coverage once prompt-injection attacks are permitted. Compared to the image-only setting ($d = 0$), certified accuracy under joint certification degrades substantially faster, despite identical image smoothing parameters. This behavior reflects the intrinsic

cost of enforcing robustness to discrete adversarial edits in addition to continuous image perturbations: certifying against even a single-token $\ell_0$ attack significantly restricts the admissible image robustness radius.

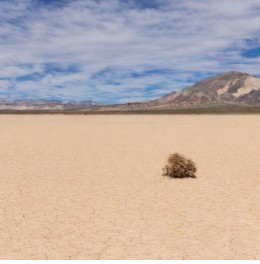

*Figure 6.* **"Look how many people love you."** Interaction-only example from the Hateful Memes dataset (Kiela et al., 2020). The image and the text are each individually classified as safe, while their joint interpretation is classified as unsafe, illustrating that harmful content can arise purely from multimodal interaction.

This behavior highlights the extreme sensitivity of certain interaction-only examples to minimal discrete perturbations under an $\ell_0$ budget constraint. In particular, a single-token edit ($d = 1$) can be sufficient to alter the joint multimodal decision and invalidate certification. Figure 6 illustrates a representative case from the Hateful Memes dataset: the image and the original text are individually safe, while their combination is classified as unsafe. Modifying a single token in the text (e.g., replacing "love" by "hate") can flip the joint prediction, making it impossible to certify robustness for the unsafe label under even a unit $\ell_0$ text budget. This illustrates that pure $\ell_0$ perturbations induce a highly fragile robustness regime in multimodal settings, where certified guarantees are intrinsically sensitive to discrete budget constraints.

### A.12. Remarks

### A.13. Computing the Hybrid Certificate for Absorbing and Uniform Text Kernels

We consider a product smoothing kernel on a discrete text modality $x_1$ and a continuous modality $x_2 \in \mathbb{R}^m$:

$$(z_1, z_2) \sim p_1(\cdot \mid x_1) \times \mathcal{N}(x_2, \sigma^2 I_m).$$

Let $p_A \in [0, 1]$ denote a one-sided lower confidence bound on the smoothed "bad" (e.g. unsafe) event, estimated by Monte-Carlo. Given a target failure level $\delta \in (0, 1)$, the hybrid certificate returns (i) a text $L_0$ radius $d^\star$ and (ii) a continuous $L_2$ radius $r^\star$.

**Text Radius.** For a fixed text kernel parameter $\beta$, compute $d^\star$ as the largest integer such that the worst-case adversarial probability under $L_0 \leq d$ is below $\delta$. For the absorbing kernel, this is available in closed form; for the uniform kernel, it is obtained by the fractional-knapsack/Neyman–Pearson bound (cf. Section A.3).

**Hybrid Radius via Neyman–Pearson.** Fix $d = d^\star$. Let the discrete likelihood ratio be

$$\gamma_1(z_1) \;=\; \frac{p_1(z_1 \mid x_1')}{p_1(z_1 \mid x_1)}, \qquad \|x_1' - x_1\|_0 \leq d,$$

and let $p_{1,\text{clean}}(z_1) = p_1(z_1 \mid x_1)$ and $p_{1,\text{adv}}(z_1) = p_1(z_1 \mid x_1')$. Define the capacity function

$$F_\sigma(t; r) \;=\; \sum_{z_1} p_{1,\text{clean}}(z_1)\, \Phi\!\left( \frac{\frac{1}{2}r^2 + \sigma^2\big(\log t - \log \gamma_1(z_1)\big)}{\sigma r} \right),$$

where $\Phi$ is the standard Gaussian CDF. For each $r > 0$, let $t^\star(r)$ be the unique solution to

$$F_\sigma\big(t^\star(r); r\big) = p_A.$$

Then the hybrid adversarial value at radius $r$ is

$$V_\sigma(r) \;=\; \sum_{z_1} p_{1,\text{adv}}(z_1)\, \Phi\!\left( \frac{\frac{1}{2}r^2 + \sigma^2\big(\log t^\star(r) - \log \gamma_1(z_1)\big)}{\sigma r} - \frac{r}{\sigma} \right).$$

The certified continuous radius is

$$r^\star = \sup\{r \geq 0 : V_\sigma(r) \leq \delta\},$$

which can be found by bisection since $r \mapsto V_\sigma(r)$ is non-decreasing.

**Absorbing Text Kernel (Closed Form $t^\star$).** For the absorbing kernel, $\gamma_1(z_1) \in \{0, 1\}$ and the discrete support reduces to two groups with masses $1 - \beta^d$ (ratio 0) and $\beta^d$ (ratio 1). Hence

$$F_\sigma(t; r) = 1 - \beta^d + \beta^d \, \Phi\left(\frac{\frac{1}{2}r^2 + \sigma^2 \log t}{\sigma r}\right),$$

so $t^\star(r)$ is obtained explicitly:

$$\Phi\left(\frac{\frac{1}{2}r^2 + \sigma^2 \log t^\star(r)}{\sigma r}\right) = \frac{p_A - (1 - \beta^d)}{\beta^d}, \quad \log t^\star(r) = \frac{\sigma r}{\sigma^2} \Phi^{-1}\left(\frac{p_A - (1 - \beta^d)}{\beta^d}\right) - \frac{r^2}{2\sigma^2}.$$

Moreover, only the ratio-1 group contributes to $V_\sigma(r)$, yielding

$$V_\sigma(r) = \beta^d \, \Phi\left(\frac{\frac{1}{2}r^2 + \sigma^2 \log t^\star(r)}{\sigma r} - \frac{r}{\sigma}\right).$$

Thus $r^\star$ is found by bisection on this scalar function.

**Uniform Text Kernel (Grouped $O(d^2)$ Computation).** For the uniform replacement kernel, $\gamma_1(z_1)$ takes multiple values, but it depends only on the pair $(i, j)$ where $i$ is the number of positions where $z_1$ differs from $x_1$ and $j$ is the number of positions where $z_1$ differs from $x_1'$ (with $0 \leq i, j \leq d$ and $i + j \geq d$). Let $V$ be the vocabulary size, $\alpha = \beta/(V - 1)$ and $\bar{\beta} = 1 - \beta$. For each feasible $(i, j)$ define the group multiplicity

$$N_{i,j} = \binom{d}{i}\binom{i}{i - (i + j - d)}(V - 2)^{i+j-d},$$

the clean group mass

$$p_{i,j} = N_{i,j} \, \alpha^i \bar{\beta}^{d-i},$$

and the likelihood ratio

$$\gamma_{i,j} = \frac{\alpha^j \bar{\beta}^{d-j}}{\alpha^i \bar{\beta}^{d-i}}.$$

After normalizing $\{p_{i,j}\}$ to sum to one, one obtains a grouped representation $\{(p_{i,j}, \gamma_{i,j})\}$ of size $O(d^2)$. Then the sums defining $F_\sigma$ and $V_\sigma$ are evaluated over groups:

$$F_\sigma(t; r) = \sum_{i,j} p_{i,j} \, \Phi\left(\frac{\frac{1}{2}r^2 + \sigma^2(\log t - \log \gamma_{i,j})}{\sigma r}\right), \qquad V_\sigma(r) = \sum_{i,j} p_{i,j}^{\text{adv}} \, \Phi\left(\frac{\frac{1}{2}r^2 + \sigma^2(\log t^\star(r) - \log \gamma_{i,j})}{\sigma r} - \frac{r}{\sigma}\right),$$

with $p_{i,j}^{\text{adv}} \propto p_{i,j}\gamma_{i,j}$. In this case $t^\star(r)$ is obtained by bisection on $t \mapsto F_\sigma(t; r)$, and $r^\star$ by bisection on $r \mapsto V_\sigma(r)$.

The grouping approach yields a computational complexity that grows quadratically with $d$ (and is independent of vocabulary size except in the binomial coefficients). For the budgets considered (up to $d = 8$), this was computationally trivial. If one needs to certify much larger $d$, additional approximation strategies or relaxations might be required, as exhaustive reasoning over extremely large text perturbations remains challenging. In practise, for safety filtering, $d = 7$ was the maximum budget with non zero certification accuracy.

