# OpenReview forum: "Certified Robustness under Heterogeneous Perturbations via Hybrid Randomized Smoothing"
_ICML.cc/2026/Conference — ICML 2026 regular_

### Official Review · Reviewer_LDaF · 2026-03-05

**Soundness:** 3
**Presentation:** 2
**Significance:** 3
**Originality:** 3
**Overall Recommendation:** 5
**Confidence:** 4

**Summary:**

The paper proposes a unified hybrid randomized smoothing framework to certify the robustness of machine learning models handling mixed discrete and continuous inputs. Existing randomized smoothing techniques treat continuous inputs (like images) and discrete inputs (like text) independently, which fails to provide sound joint theoretical guarantees when adversaries perturb both modalities simultaneously. The authors demonstrate that introducing continuous Gaussian noise effectively regularizes the discrete likelihood ratio. This resolves the intractable fractional knapsack problem typically associated with discrete smoothing, resulting in a one-dimensional, invertible likelihood-ratio cumulative distribution function. The methodology is validated on a tabular dataset and a multimodal safety filtering task using LLaVA-Guard, successfully demonstrating valid joint robustness certificates against heterogeneous perturbations.

**Compliance With Llm Reviewing Policy:**

Affirmed.

**Final Justification:**

The paper proposes a mathematically sound theoretical framework for joint randomized smoothing across mixed inputs. In my initial review, I acknowledged the theoretical contribution but raised concerns regarding computational overhead, semantic preservation under uniform kernels, and the clarity of Figures 1 and 3.

The authors provided a highly factual and constructive rebuttal. They presented concrete empirical evidence of an 8x inference speedup, which sufficiently mitigates the computational bottlenecks for practical deployment. Furthermore, the newly provided comparative data clearly isolates the semantic impact, demonstrating that the suffix-only setting preserves accuracy significantly better than full-text corruption. The authors also committed to the necessary visual revisions.

Because the rebuttal comprehensively addressed all my main concerns with solid quantitative data, my assessment has positively changed. I have raised my score accordingly and recommend accepting this paper, provided the new evaluations and clarified figures are integrated into the final manuscript.

**Key Questions For Authors:**

1.Could you redesign or visually differentiate Figure 1 and Figure 3? Making their distinct conceptual purposes (vulnerability concept vs. empirical attack setup) immediately clear would greatly improve the readability and prevent potential reader confusion.

2.Regarding the substantial computational cost, are there any heuristic approximations, adaptive sampling methods, or early-stopping criteria that could reduce the certification time in practice without completely sacrificing the conservative theoretical guarantees?

3.For the uniform replacement kernel, how does the choice of vocabulary size and corruption rate practically impact the semantic integrity of the prompt? Does the base model often misclassify clean but uniformly corrupted text during the Monte Carlo sampling phase?

**Limitations:**

Yes. The authors adequately discuss the limitations regarding the computational cost and the degeneracy of absorbing kernels in the appendix and experimental sections.

**Strengths And Weaknesses:**

Strengths:

1.The motivation is highly relevant and timely. With the rapid deployment of multimodal foundation models, joint adversarial attacks across different modalities represent a realistic and critical threat. Identifying and addressing the non-composability of unimodal robustness certificates is a significant contribution to the field of AI safety.

2.The theoretical framework is mathematically elegant and sound. Leveraging continuous Gaussian noise to smooth the step-functions of discrete likelihood ratios is a clever insight. The derivation that reduces a complex joint optimization challenge into a tractable 1D root-finding problem successfully unifies previous continuous and discrete randomized smoothing limits.

3.The experimental design effectively isolates the problem. Evaluating the framework on an interaction-only dataset where individual modalities are safe but their combination is unsafe perfectly demonstrates the necessity and soundness of the proposed joint certification approach.

Weaknesses:

1.The visual presentation of the manuscript could be significantly enhanced. The paper contains very few figures, and the existing diagrams are somewhat rudimentary and potentially misleading. Specifically, Figure 1 and Figure 3 share a highly similar visual structure but serve entirely different logical purposes. Figure 1 illustrates an interaction-only vulnerability concept, whereas Figure 3 depicts the specific threat model and empirical attack mechanism. This visual overlap risks confusing readers who are scanning the paper for context.

2.The computational overhead of the proposed hybrid certification is substantial. As acknowledged by the authors, computing the certificate for a single datapoint takes approximately 500 seconds. This high cost might limit the scalability of the method in real-time or large-scale deployment scenarios compared to standard image-only smoothing.

3.The reliance on uniform replacement kernels for text, necessary to avoid the exponential decay associated with absorbing kernels under suffix attacks, might constrain the semantic preservation of the textual input in broader natural language tasks outside of safety filtering.

---

> ### Author Rebuttal · Authors · 2026-03-27
>
> We thank the reviewer for recognizing the timely motivation (S1), mathematical elegance (S2), and effective experimental isolation of the interaction-only problem (S3).
>
> **W1 / Q1: Visual similarity between Figure 1 and Figure 3.** We agree. Figure 1 will be reworked as a conceptual/schematic illustration (e.g., decision-space diagram) emphasizing the logical point that joint unsafety cannot be inferred from unimodal safety. Figure 3 will adopt a visually distinct flow-diagram style with explicit attack arrows. The two figures will be immediately distinguishable.
>
> **W2 / Q2: Computational cost and speedups.** We agree that the computational overhead is substantial. The dominant cost is repeated VLM forward passes; the certificate computation itself (NP root-finding, discrete aggregation) adds <1s per sample. Relative to image-only RS, one structural source of additional cost is that hybrid certification computes a discrete--continuous frontier in $(d,\varepsilon)$, rather than a single scalar certificate.
>
> We implemented two concrete speedups in our current pipeline:
>
> 1. Optimized batching and FlashAttention yield an approximately 30\% inference speedup in our setup;
>
> 2. As an efficiency–accuracy tradeoff, a one-shot shared-$d$ variant for suffix attacks reuses a single MC estimate across all discrete budgets, providing an 8× speedup (from ~500s to ~44s per sample for $d_{\max}=8$) with moderate certificate loss (mean radius 2.07→1.55, mean text budget 1.48→1.23). For $\ell_0$ attacks, certification is already one-shot.
>
> More broadly, confidence-sequence-based early stopping (Voráček, 2024) and input-adaptive sampling (Chen et al., 2022) are promising directions for further sample reduction, with recent work showing 1–2 orders of magnitude sample reduction with moderate radius degradation (Seferis et al., 2024; 2025). We will discuss these tradeoffs in the revision.
>
> **W3: Semantic preservation under uniform kernels.** In the suffix setting, the uniform kernel corrupts only appended tokens; the core prompt remains intact, so the primary semantic content is much better preserved than in full-text smoothing. For broader NLP tasks where the entire text is smoothed, we agree that uniform replacement may be less well suited, and combining discrete smoothing with text denoising mechanisms (analogous to the image diffusion denoiser) is a concrete future direction, though it would require a separate certification analysis.
>
> **Q3: Vocabulary size and corruption rate.** For vocabularies of size $|V| \approx 32{,}000$, each corrupted token is effectively replaced by a nearly uniform random token ($\alpha = \beta/(|V|-1)$). The practical impact depends strongly on the threat model.
>
> On 100 initially well-classified interaction-only examples ($\sigma=0.5$), smoothed accuracy under full-text $\ell_0$ corruption drops sharply: 74.76% at $\beta=0.1$, 33.81% at $\beta=0.25$, 2.86% at $\beta=0.5$. By contrast, under suffix attacks where only appended tokens are corrupted, smoothed accuracy remains high: 95.71% at $\beta=0.1$, 93.99% at $\beta=0.25$, 88.41% at $\beta=0.5$.
>
> | Threat model | $\beta=0.1$ | $\beta=0.25$ | $\beta=0.5$ |
> |---|---:|---:|---:|
> | $\ell_0$ (full-text) | 74.76% | 33.81% | 2.86% |
> | Suffix (append-only) | 95.71% | 93.99% | 88.41% |
>
> This confirms that the suffix setting is far more favorable for uniform-kernel RS, since corruption is localized to appended suffix tokens. Higher smoothed accuracy directly translates to higher certification coverage. We will add this table in the revision.
>
> ---
>
> Voracek, Treatment of Statistical Estimation Problems in Randomized Smoothing for Adversarial Robustness. (NeurIPS), 2024
>
> Chen et al., Input-Specific Robustness Certification for Randomized Smoothing. (AAAI), 2022
>
> Seferis et al., Estimating the Robustness Radius for Randomized Smoothing with 100× Sample Efficiency.  (ECAI), 2024.
>
> Seferis et al., Randomized Smoothing Meets Vision-Language Models. (EMNLP), 2025.

---

> > ### Author Rebuttal · Reviewer_LDaF · 2026-04-02
> >
> > Thank you for the thorough rebuttal. The authors have rigorously addressed all my concerns with concrete empirical evidence rather than just theoretical promises. Specifically, the demonstrated 8x inference speedup using the one-shot shared variant adequately resolves my concerns regarding the computational bottleneck, even with the acknowledged moderate certificate loss. Furthermore, the newly provided table clearly and quantitatively isolates the semantic impact of the uniform kernel, successfully proving that the suffix-only setting preserves accuracy significantly better than full-text corruption. The commitment to visually differentiating Figure 1 and Figure 3 is also noted and satisfactory. My concerns are fully resolved, and I strongly encourage the authors to integrate all these new speedup evaluations and empirical tables into the final manuscript.

---

> > > ### Author Response · Authors · 2026-04-02
> > >
> > > We thank the reviewer for the follow-up and will integrate these changes.

---

### Official Review · Reviewer_c6zN · 2026-03-10

**Soundness:** 3
**Presentation:** 4
**Significance:** 3
**Originality:** 2
**Overall Recommendation:** 4
**Confidence:** 3

**Summary:**

In this work, an unified randomized smoothing approach for continuous and discrete modalities is presented. The work carefully addresses the gap in the literature where perturbations in different modalities are treated separately.

**Compliance With Llm Reviewing Policy:**

Affirmed.

**Final Justification:**

Authors' rebuttal addressed my concerns. I recommend accepting the paper.

**Key Questions For Authors:**

Please comment on the weaknesses above.

**Limitations:**

Computation complexity and expected performance degradation are the main limitations for practical applications.
Please use the evaluation sparingly.

**Strengths And Weaknesses:**

S:


1) The paper proposed a theoretical grounding for the unified randomized smoothing. The proof seems correct to me
2) The work is well-written and the problem is well-motivated.
3) The experimental evaluation is rigorous (importantly, an evaluation against an adaptive attack is presented)


W:

1) The work would benefit from the comparison with the other multimodal certified robustness methods, for example, I) and II)
2) Like other smoothing-based approaches, the computation complexity is a major bottleneck (the inference of a single object takes ~500s on a H100) what limits the applicability of the method.
3) Applying randomized smoothing-based techniques expectedly leads to a performance degradation; this degradation is not reflected in the manuscript and should be added.


I) Sun, A., Ma, C., Ge, K., & Vosoughi, S. (2024). Achieving domain-independent certified robustness via knowledge continuity. Advances in Neural Information Processing Systems, 37, 49552-49590.
II) Wang, Y., Fu, H., Zou, W., & Jia, J. (2024). Mmcert: Provable defense against adversarial attacks to multi-modal models. In Proceedings of the IEEE/CVF Conference on Computer Vision and Pattern Recognition (pp. 24655-24664).

---

> ### Author Rebuttal · Authors · 2026-03-27
>
> We thank the reviewer for confirming correct proofs (S1), clear writing (S2), and rigorous evaluation including adaptive attacks (S3).
>
> **W1: Comparison with MMCert and Knowledge Continuity.** We will add a discussion in Section 2.
>
> *MMCert (Wang et al., 2024)* certifies robustness via independent subsampling of discrete elements across modalities, followed by majority voting. Its perturbation model is $\ell_0$ per modality (modify/add/delete discrete elements), and the certificate counts overlaps between subsampled sets. This differs from our setting: we handle heterogeneous perturbation geometry ($\ell_0$ discrete + $\ell_2$ continuous), whereas MMCert treats all modalities as bags of discrete elements. MMCert's subsampling also cannot certify the interaction-only regime (Figure 1): ablating a modality removes the cross-modal interaction producing the unsafe prediction.
>
> *Knowledge Continuity (Sun et al., 2024)* bounds a volatility measure (change in loss relative to representation-space distance) via concentration inequalities. The guarantees are probabilistic and distributional, not worst-case over a perturbation budget.
> The guarantee is distributional rather than a worst-case budget certificate, a fundamentally different paradigm from RS not directly applicable to our joint ($\ell_0$, $\ell_2$) threat model.
>
> To our knowledge, our framework is the first joint Neyman–Pearson certificate under heterogeneous perturbation models. A direct numerical comparison is not straightforward because the perturbation models do not overlap, but we will discuss both methods in the revised Section 2.
>
> **W2: Computational complexity.** We agree that computational cost is a practical limitation shared by all RS methods applied to large VLMs. The ~500s per sample is dominated by VLM inference (LLaVA-Guard); the hybrid certificate computation itself (NP root-finding, discrete aggregation) adds <1s. This cost is comparable to text-only RS (Table 3); the certificate-side computation is negligible relative to repeated VLM inference.
>
> We implemented two concrete speedups in our current pipeline:
> 1. Optimized batching and FlashAttention yield an approximately 30\% inference speedup in our setup.
> 2. As an explicit efficiency--accuracy tradeoff, we evaluated a one-shot shared-$d$ variant that reuses a single Monte Carlo estimate across all discrete budgets, reducing runtime from about 500s to 44s per sample (roughly $8\times$ for $d_{\max}=8$) at the cost of weaker certificates (mean radius $2.07 \to 1.55$, mean certified text budget $1.48 \to 1.23$). For $\ell_0$ attacks, certification is already one-shot.
>
> More broadly, confidence-sequence-based early stopping (Voráček, 2024) and input-adaptive sampling (Chen et al., 2022) are promising directions for further sample reduction. We will discuss these efficiency–accuracy tradeoffs in the revision.
>
> **W3: Performance degradation.** We agree and report smoothed accuracy below. On 100 interaction-only examples classified Unsafe at ($\sigma=0$, $\beta=0$) by LLaVA-Guard on Hateful Memes:
>
> **$\ell_0$ threat model — smoothed accuracy:**
>
> | $\sigma$ | $\beta=0$ | $\beta=0.1$ | $\beta=0.25$ | $\beta=0.5$ |
> |---|---:|---:|---:|---:|
> | 0.0 | 100.00% | 75.71% | 32.38% | 3.81% |
> | 0.5 | 91.90% | 74.76% | 33.81% | 2.86% |
> | 1.0 | 88.57% | 68.10% | 32.86% | 2.38% |
>
> **Suffix threat model — smoothed accuracy:**
>
> | $\sigma$ | $\beta=0$ | $\beta=0.1$ | $\beta=0.25$ | $\beta=0.5$ |
> |---|---:|---:|---:|---:|
> | 0.0 | 100.00% | 100.00% | 94.85% | 88.84% |
> | 0.5 | 95.28% | 95.71% | 93.99% | 88.41% |
> | 1.0 | 91.85% | 91.85% | 90.99% | 84.98% |
>
> The dominant source of utility degradation is text corruption: under $\ell_0$, $\beta=0.25$ reduces accuracy from 100% to approximately 33%; under suffix attacks, the cost is moderate ( approximately 94%) since only appended tokens are corrupted. Image smoothing introduces minimal additional degradation across all $\sigma$ values. This resilience to increasing $\sigma$ is a direct consequence of the diffusion denoiser used in our image pipeline, following the denoised RS paradigm of Carlini et al. (2023b): the denoiser projects noisy images back toward the data manifold, effectively absorbing the injected Gaussian noise. Extended $\sigma$ sweeps at $\beta=0.25$ confirm this:
>
> | $\sigma$ | $\ell_0$ smoothed acc. | Suffix smoothed acc. |
> |---|---:|---:|
> | 0.0 | 32.38% | 94.85% |
> | 0.5 | 33.81% | 93.99% |
> | 1.0 | 32.86% | 90.99% |
> | 2.0 | 26.67% | 75.54% |
> | 3.0 | 24.29% | 67.38% |
> | 5.0 | 15.24% | 48.07% |
>
> Smoothed accuracy remains stable up to $\sigma \approx 1.0$ and degrades beyond. In practice, our certificates use $\sigma \in \{0.5, 1.0\}$, well within this stable regime. We will include these tables alongside certified accuracy in the revision.
>
>
> Voracek, Treatment of Statistical Estimation Problems in Randomized Smoothing for Adversarial Robustness. (NeurIPS), 2024.
>
> Chen et al, Input-Specific Robustness Certification for Randomized Smoothing. (AAAI), 2022.

---

> > ### Author Rebuttal · Reviewer_c6zN · 2026-04-01
> >
> > I thank authors for their rebuttal; however, I would still consider adding comparison with the multimodal certified robustness methods to further illustrate experimentally the benefits and limitations of proposed approach.

---

> > > ### Author Response · Authors · 2026-04-02
> > >
> > > We thank the reviewer for the follow-up and have carried out additional experimental and analytical work to address this point.
> > >
> > > **1. Experimental comparison with MMCert-style subsampling on Hateful Memes.**
> > > Following the reviewer's suggestion, we implemented an MMCert-style independent subsampling baseline on our Hateful Memes evaluation set using the same safety detector (LLaVA-Guard). We restrict this comparison to a subset of approximately 100 examples that are classified as Unsafe without ablation, in order to isolate whether certification survives partial observation once the base prediction is correct. We treated text tokens and $16 \times 16$ image patches as basic elements and estimated the smoothed unsafe probability $p_A$ from $N=100$ Monte Carlo samples per configuration. Since this is a binary classification setting, the relevant diagnostics are the number of examples with $\underline{p}_A > \tfrac{1}{2}$ together with the mean $\hat{p}_A$:
> > >
> > > | Keep ratio $(k_1/n_1 = k_2/n_2)$ | Examples with $\underline{p}_A > 0.5$ | Mean $\hat{p}_A$ |
> > > |---|---:|---:|
> > > | 1.0 (no ablation) | 100 / 100 | 1.0 |
> > > | 0.50 | 0 / 100 | 0.120 |
> > > | 0.20 | 0 / 100 | 0.056 |
> > > | 0.10 | 0 / 100 | 0.033 |
> > > | 0.05 | 0 / 100 | 0.018 |
> > >
> > > Under every ablated keep ratio, MMCert-style subsampling yields zero certifiable examples. The mean $\hat{p}_A$ values confirm that this is not a borderline failure: the smoothed probability collapses far below the $\tfrac{1}{2}$ threshold required for a non-trivial certificate in this binary setting.
> > >
> > > We further verified that the choice of patch granularity does not alter the conclusion. Repeating the diagonal sweep with $32 \times 32$ patches yields mean $\hat{p}_A$ of 0.009 at $k/n=0.05$, 0.018 at $0.1$, and 0.039 at $0.2$, slightly lower than with $16 \times 16$ patches. The collapse is therefore not a patch-size artifact.
> > >
> > > We also tested asymmetric allocations of the keep budget across modalities. These did not change the overall conclusion: keeping the full image but only $5\%$ of text yields $0/100$ certifiable examples, while keeping the full text but only $5\%$ of image yields $25/100$. This indicates that, in interaction-dependent settings, preserving a single modality is insufficient: removing most of the other modality can destroy the cross-modal signal required for correct prediction.
> > >
> > > This failure is structural. MMCert relies on prediction stability under partial observation, which is well matched to tasks with redundant evidence spread across tokens, patches, or frames. In contrast, the unsafe label in Hateful Memes often depends on sparse cross-modal alignment between specific textual and visual cues. Independent subsampling can therefore remove a critical part of the joint signal even when much of each modality is retained, causing the smoothed unsafe probability to collapse far below the level needed for certification. Intuitively, the probability of preserving all jointly necessary cues decays multiplicatively with subsampling across modalities.
> > >
> > > By contrast, as shown in our initial rebuttal tables and paper, Hybrid RS on the same Hateful Memes setting maintains non-trivial smoothed accuracy and certification under joint $(d,\varepsilon)$ perturbations, precisely because noise injection preserves the cross-modal structure that subsampling destroys.
> > >
> > > **2. Positioning relative to Knowledge Continuity.**
> > > As discussed in our previous rebuttal, Knowledge Continuity provides a distributional representation-space guarantee rather than a worst-case certificate under an explicit perturbation budget, so we position it as complementary rather than as a directly comparable baseline. We clarify this distinction in the revised Section 2.
> > >
> > > **3. Structured comparison table (revised Section 2).**
> > >
> > > |  | MMCert | Knowledge Continuity | Ours |
> > > |---|---|---|---|
> > > | Perturbation model | $\ell_0$ per modality (discrete) | Representation-space | $\ell_0$ discrete + $\ell_2$ continuous |
> > > | Guarantee | Worst-case (NP, subsampling) | Probabilistic, distributional | Worst-case (hybrid NP) |
> > > | Heterogeneous inputs | No (homogeneous discrete) | N/A (unimodal) | Yes |
> > > | Cross-modal interaction | Weak in interaction-dependent regimes | Not evaluated | Yes (noise preserves signal) |
> > > | Favorable regime | Redundant, set-like signals | — | Sparse, compositional, cross-modal |
> > >
> > > We will incorporate the experimental results, asymmetric sweep analysis, and comparative discussion in the revised manuscript.

---

### Official Review · Reviewer_6K8h · 2026-03-13

**Soundness:** 3
**Presentation:** 3
**Significance:** 3
**Originality:** 4
**Overall Recommendation:** 4
**Confidence:** 4

**Summary:**

This paper novelly proposes a cross-modal randomized smoothing framework that integrates discrete and continuous variables, and derives the corresponding method for computing certificates. The authors also validate their theory on multimodal data.

**Compliance With Llm Reviewing Policy:**

Affirmed.

**Final Justification:**

The paper is well-motivated and addresses the highly popular multimodal setting. To my knowledge, it is the first to provide robustness certificates for both discrete and continuous perturbations simultaneously, which is quite novel. Furthermore, the rebuttal clearly clarified that the work is built upon a classic paper in the field, resolving my concerns from the initial review phase. Consequently, I have increased the Soundness score from 1 to 3, Originality from 3 to 4, and the overall Rating from 3 to 4. Overall, I believe this is a solid paper.

**Key Questions For Authors:**

See S&W

**Limitations:**

yes

**Strengths And Weaknesses:**

Strengths:

(1) This work points out the shortcomings of trivially combining different modalities to obtain certificates, and unifies multimodal adversarial perturbations into a single robustness guarantee. This provides strong motivation for the paper.

(2) The paper integrates discrete and continuous distributions and transforms them into a new continuous distribution. This distribution has a monotonicity property that enables certificate computation, which is a valuable finding.

Weaknesses:

(1) The robustness certification provided by this method does not outperform that of the pure-text setting, and is also weaker than that of the pure-image setting, which reduces the contribution of this work.

(2) This method is computationally very expensive. For a single sample, certification with 10,000 Monte Carlo samples takes 500 seconds, yet the resulting confidence is still not high. In essence, mixing modalities does not improve the practicality of randomized smoothing. Similarly, in Table 4, the number of samples is only 1,000. Does this mean the confidence level is only 90%? If so, the reliability of these results is too low.

(3) The evaluation is not comprehensive, possibly due to the high experimental cost. The datasets used in the paper are all too small, and even for the same task, the variety of datasets is too limited.

(4) A very important issue is that one of the paper’s main references, Chen et al., 2025a, which models the discrete case as a knapsack problem, is still only an arXiv paper and has not passed peer review. This significantly reduces confidence in the present work. Given this, I tend to remain conservative and believe that it may be too early to accept this paper.

---

> ### Author Rebuttal · Authors · 2026-03-27
>
> We thank the reviewer for the constructive review.
>
> **W1: "Hybrid does not outperform unimodal."** This comparison should be interpreted carefully, because the certified threat models differ. The hybrid certificate certifies stability under *joint* discrete–continuous perturbations, a strictly stronger adversary. Proposition 4.1 shows that composing unimodal certificates is provably unsound. By construction, the hybrid radius at $d > 0$ must be at most the image-only radius at $d = 0$, because the adversary is more powerful.
> The relevant baseline is "a sound joint guarantee vs. no guarantee at all" in the interaction-only regime where unimodal certificates do not apply (Figure 1). Reduction is limited: the hybrid image radius ($\bar{r} = 3.76$ at $d = 1$) is within 6% of image-only ($\bar{r} = 3.99$ at $d = 0$).
>
> The contribution is a sound joint guarantee in a regime where none existed; the modest radius reduction is the expected cost of a strictly stronger adversary.
>
> **W2: Computational cost.** The ~500s per sample is dominated by repeated VLM inference (LLaVA-Guard); the certificate-side computation itself (NP root-finding and discrete aggregation) adds <1s. Relative to image-only RS, the additional runtime mainly comes from evaluating multiple discrete budgets up to $d_{\max}$, since hybrid RS certifies a frontier in the $(d, \varepsilon)$ plane rather than a single scalar radius.
>
> We implemented two concrete speedups. First, optimized batching and FlashAttention yield an approximately 30% inference speedup. Second, a one-shot shared-$d$ variant for suffix attacks reduces runtime by about $8\times$ (~500s $\to$ ~44s per sample for $d_{\max}=8$) with moderate certificate degradation (mean radius $2.07 \to 1.55$, mean certified text budget $1.48 \to 1.23$).
>
> For $\ell_0$ text, certification is already one-shot, so this speedup can be obtained without certificate degradation  (~44s per sample). Additional reductions may be possible via confidence-sequence early stopping (Voráček, 2024) and input-adaptive sampling (Chen et al., 2022).
>
> Certification uses $n=10{,}000$ Monte Carlo samples with a one-sided Clopper–Pearson bound at $\alpha=0.01$; yielding a conservative guarantee by construction. In our setting, the main practical bottleneck is the runtime from repeated VLM inference.
>
> The dominant cost is VLM inference; the certificate-side post-processing is negligible (< 1s per sample).
>
> **Confidence in Table 4.** Table 4 reports *empirical attack results*, not certification. The $n = 1{,}000$ samples estimate the smoothed "Safe" probability *within the attack*, as a measure of attack strength (the attacker's goal is Safe% $> 1 - \tau \approx 99.995\%$). Certification (Table 1, Figure 4) uses $n = 10{,}000$ with Clopper–Pearson at $\alpha = 0.01$. The observed Safe% values (74–90%) are far from the 99.995% threshold needed to violate the certificate, so the conclusion is robust to sample size. We will clarify this distinction in the revision.
>
> **W3: Limited datasets.** We agree that a broader evaluation would strengthen the empirical evidence. Our current experiments isolate the interaction-dependent regime, which requires careful curation: removing unimodal-unsafe examples and verifying that unsafe behavior arises from modal interactions. We also include a tabular experiment (ADULT) to demonstrate generality beyond vision–language settings.
>
> We additionally ran the full pipeline on MM-SafetyBench (1680 samples). Most samples are unimodally safe or unimodally unsafe. Only 7.5% are interaction-only failures, i.e., text-safe and image-safe individually yet unsafe jointly, which is consistent with MM-SafetyBench not being curated for this regime. In this subset, the mean certified text budget is 3.62 and the mean certified radius is 3.37. These additional runs show that the pipeline transfers beyond our curated split without retuning, although coverage remains limited because the benchmark is not designed for interaction-only failures.
>
> Interaction-only evaluation is inherently constrained by available benchmarks; MM-SafetyBench confirms the pipeline transfers without retuning.
>
> **W4: Chen et al. is only arXiv.** Theorem 4.2 is self-contained: the proof relies only on the Neyman–Pearson lemma (1933), Gaussian density properties, and convex order arguments. The knapsack interpretation (Section 3) is a direct reformulation of the likelihood-ratio greedy procedure in Lee et al. (2019, Lemma 2); Chen et al. is cited for terminology only.
>
> No step of Theorem 4.2 depends on Chen et al.; the proof is self-contained, and we will revise Section 3 to make this explicit relative to Lee et al.
>
> ---
> Liu et al., MM-SafetyBench: A Benchmark for Safety Evaluation of Multimodal Large Language Models. (ECCV), 2024
>
> Voracek, Treatment of Statistical Estimation Problems in Randomized Smoothing for Adversarial Robustness. (NeurIPS), 2024
>
> Chen et al., Input-Specific Robustness Certification for Randomized Smoothing. (AAAI), 2022

---

> > ### Author Rebuttal · Reviewer_6K8h · 2026-04-03
> >
> > I thank the authors for their rebuttal. It has addressed my concerns and cleared up my misunderstandings. I have decided to increase my score.

---

### Decision · Program_Chairs · 2026-04-30

**Decision:**

Accept (regular)

**Comment:**

Thanks for the submission to ICML 2026. The submission extends randomized smoothing for mixed discrete-continuous inputs that improve multimodal model safety certifiably against joint text-image adversarial attacks. All reviewers appreciated the strength of the submission in providing theoretically rigorous robustness guarantees for multimodal foundation models, which is a timly topic, with rigorous evaluation. All reviewers and authors were actively involved in discussion phase. After discussion, all reviewers labelled that their concerns (computational overhead of randomized smoothing, performance degradation, and comparison with other multimodal certified defenses) were resolved and recommended accptance. Though some key issues still persist such as the computational overhead and performance degradation, these also exist for other randomized smoothing techniques and would be future avenues. Hence, as an AC I recommend acceptance as well should space permit. Please incorporate the reviewer feedback in camera-ready version.